# Multiplex base editing to convert TAG into TAA codons in the human genome

Yuting Chen[1,2,3,5], Eriona Hysolli [1,2,5✉], Anlu Chen[4,5], Stephen Casper[1,5], Songlei Liu [1,2], Kevin Yang[1], Chenli Liu [3✉] & George Church [1,2✉]

Whole-genome recoding has been shown to enable nonstandard amino acids, biocontainment and viral resistance in bacteria. Here we take the first steps to extend this to human cells demonstrating exceptional base editing to convert TAG to TAA for 33 essential genes via a single transfection, and examine base-editing genome-wide (observing ~40 C-to-T off-target events in essential gene exons). We also introduce GRIT, a computational tool for recoding. This demonstrates the feasibility of recoding, and highly multiplex editing in mammalian cells.

[1] Department of Genetics, Harvard Medical School, Boston, MA 02115, USA. [2] Wyss Institute for Biologically Inspired Engineering, Boston, MA 02115, USA. [3] CAS Key Laboratory of Quantitative Engineering Biology, Center for Genome Engineering and Therapy, Shenzhen Institute of Synthetic Biology, Shenzhen Institute of Advanced Technology, Chinese Academy of Sciences, Shenzhen 518055, China. [4] Division of Endocrinology, Diabetes and Metabolism, Beth Israel Deaconess Medical Center, Harvard Medical School, Boston, MA 02215, USA. [5]These authors contributed equally: Yuting Chen, Eriona Hysolli, Anlu Chen, Stephen Casper. ✉email: eriona.hysolli@gmail.com; cl.liu@siat.ac.cn; gchurch@genetics.med.harvard.edu

The genetic code is degenerate, assigning 61 triplet codons to 20 naturally occurring amino acids in addition to 3 triplets coding for a stop signal, and 18 of the 20 amino acids are encoded by more than one synonymous codon[1]. Genome recoding is a powerful tool to understand and enhance the genomic function of organisms by genetic engineering. Recoding confers virus resistance[2–5], and can also be repurposed to assign the "blank" codons new functions including nonstandard amino acid incorporation[2] and biocontainment[6,7]. Recoding was first established in prokaryotes through substitution of the TAG stop codon with TAA and deletion of release factor 1 (RF1)[2,8]. Recently, recoding was implemented genome-wide in E. coli by replacing two sense codons with their synonymous codons, and deleting the corresponding transfer RNA (tRNA)[3]. Then, recoding has also been subsequently extended to yeast genome[9], but its application in the human genome has not been reported so far. Here, we propose human genome recoding to generate virus-resistant cell lines by converting stop codon TAG to TAA, and replacing the endogenous eukaryotic release factor 1 (eRF1) with engineered eRF1 variants (Supplementary Fig. 1a). Human recoding is also the pilot project of GP-write, which was founded to evolve the "reading" goals of the Human Genome Project into "writing" the next generation genomes[10].

Our lab first achieved genome-wide recoding where 314 instances of the UAG stop codon were replaced with UAA in E. coli[8]. Virus resistance was subsequently tested in engineered E. coli with all UAG to UAA replacements and deleted RF1 that enables the termination of translation for UAG and UAA[11]. This recoding scheme decreased transduction by 4 different bacteriophages (λ, M13, P1, MS2) that infect E. coli[2,4,5]. In another effort, rewriting of 13 sense codons across a set of ribosomal genes[12] and 123 instances of two rare Arginine codons were synonymously replaced[13]. Recently, our lab has made over 62,214 changes by synthesizing and assembling a 3.97-megabase, 57-codon E. coli genome[14]. Parallel efforts have resulted in the complete recoding and assembly of a 61-codon E. coli strain[15]. Deletion of the tRNAs charging the removed serine codons and release factor 1 conferred resistance from a cocktail of viruses, and the blank codons were reassigned to enable the efficient synthesis of proteins containing three distinct nonstandard amino acids in SYN61[3]. In addition to E.coli, 1557 synonymous leucine codons were replaced across 176 genes in Salmonella typhimurium using SIRCAS[16]. Redesigning and de novo synthesis of yeast genomes project was implemented by the SC 2.0 consortium team, and UAG to UAA recoding in their design[9,17,18].

Building on this previous work, we set out to explore the feasibility of genome-wide TAG to TAA replacement in human cells. We selected amber stop code TAG for the following reasons: (1) Previously published papers reported that recoded E. coli showing nonstandard amino acids incorporation and multiple viruses resistance[2,3]; (2) TAG is the least commonly used codon in the human genome that allows for fewer edits; (3) TAG could be theoretically edited to TAA using C to T base editors (CBE)[19], and increase flexibility in gRNAs design as TAG denotes the end of the gene, thus reducing concern for CBEs-induced bystander edits[20] effects on gene transcription and translation.

## Results

**Software design for human genome recoding.** Given the scale of genome-level recoding in human cells, there is a need for software that can automate the process of part design. To meet this need, we designed Genome Recoding Informatics Toolbox (GRIT), which provides a python-based platform for genome-scale data analysis tailored to recoding (Fig. 1a). We use the acronym GRIT to reflect the perseverance required for genome-scale engineering.

The central functions of GRIT are to parse genome data, find TAG codons, and identify guides for base editors with NG protospacer adjacent motifs (PAMs)[21].

GRIT offers a toolkit for informatics with an emphasis on recoding. It was created with three key design principles: (1) Portability: all data can be downloaded, and GRIT can be run from a desktop computer in minutes. (2) Adaptability: the full source for the project is in two python files, and a diversity of general and recoding-specific informatics data are readily available including full gene and chromosome sequences. (3) Ease of use: GRIT comes with prewritten methods for replicating results and analyzing chromosome data, gene data, TAG site data, and guides for editing. For recoding in particular, GRIT can be used to index all TAG sites in the genome, search for ones that can be directly edited with a C base editor or edited with a "daisy" chain of A and C editors, and to design the corresponding guides.

GRIT works by creating chromosome and gene objects where each store bioinformatic data. For chromosomes, GRIT gathers data including chromosome name, wildtype sequence, recoded sequence, indices of sites to recode, base editor sites, gene objects, and edit sites that are part of different genes or different codons read in different frames and which two genes they are part of. For genes, GRIT stores the gene name, chromosome, strand, wildtype sequence, recoded sequence, active isoform, introns, isoform information, gene essentiality data, and recoding sites. By making this data readily available, GRIT can be easily adapted for purposes beyond recoding.

GRIT relies on human genome sequence data from GRCh38.p13 and gene essentiality data from the OGEE database[22]. Using these data, GRIT identifies 6700 total TAG sites (including ones in alternate isoforms of the same genes). Of the genes that contain TAG sites, 5266 have one isoform, 574 have two isoforms, 80 have three isoforms, 9 have four isoforms, and 2 have five isoforms. Of these, 6648 are editable across the human haploid genome using base editors with editing window from position 1-13[23]. Additionally, 1947 (1937 of which are editable) of the 6700 TAG codons are in genes that have evidence of essentiality (Fig. 1b). Using GRIT, we also visualized the distribution of TAG sites throughout the 24 human chromosomes (Fig. 1c). In addition to core functions related to recoding, GRIT can be used more generally for informatics involving coding DNA sequences, chromosomal sequences, gene essentiality, multiple isoforms, multifunctional sites, gRNAs, and primers.

**Multiplexed base editing in HEK293T cells by gRNA arrays.** CRISPR/Cas9[24–26], base editors[19,27] and a prime editor[28] have greatly accelerated the speed of genome engineering[29]. Although CBE can be used for TAG to TAA conversion with suitable gRNA design, existing multiplexed gene editing technologies[30] do not meet the demand, we need to develop technologies that simultaneously deliver multiple gRNAs and base editors protein into a single mammalian cell for TAG to TAA recoding. With the advancements in DNA synthesis capability and exponential cost reduction, we directly designed and synthesized gBlocks containing five individual gRNA cassettes: five previously published sgRNAs[23] (gBlock-PC) and five designed sgRNAs targeting TAG regions of genes (gBlock-YC1) (Fig. 2a). We transiently co-transfected gBlock-PC and gBlock-YC1 separately with evoAPOBEC1-BE4max-NG[23] into HEK293T cells. Sanger sequencing and EditR[31] analysis showed that the efficiency of sgRNAs from gBlock-PC is ~40–50%, which is slightly lower than those with the same sgRNAs delivered individually[23], and the efficiency of sgRNAs from gBlock-YC1 is ~20–50% (Fig. 2b, c). Then, utilizing piggybac transposon system, we generated two stable and doxycycline-inducible HEK293T lines

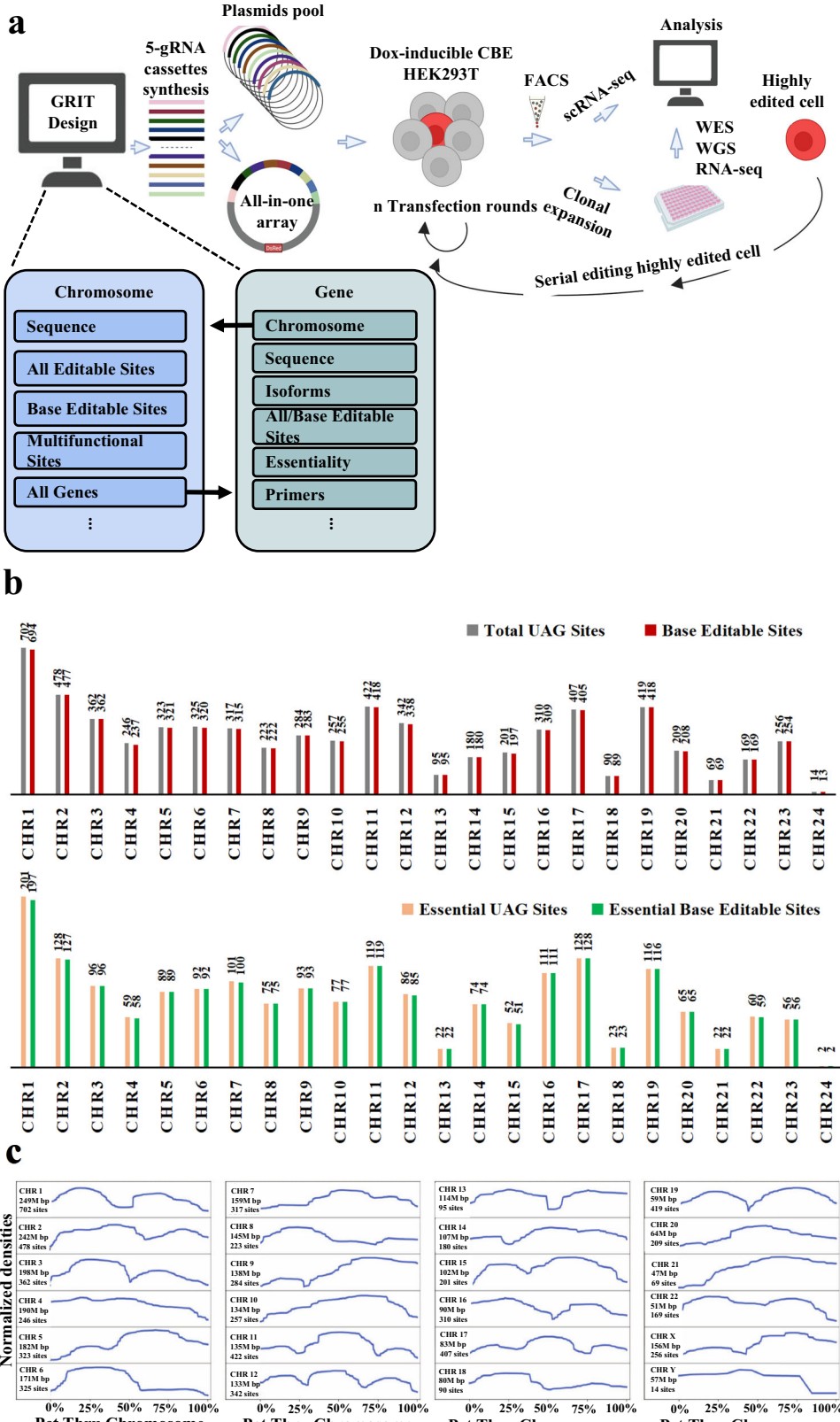

**Fig. 1 Software GRIT and framework for converting UAG to UAA for human recoding. a** Framework for converting TAG codons into TAA in human cells. Chromosome and gene objects structure in GRIT. Using GRIT for informatics centers around chromosome and gene objects. Each box contains some (but not all) of the attributes for each. Each chromosome contains a list of gene objects, and each gene object contains its corresponding chromosome's object. $n = 2$ in our current experiment. CBE, cytosine base editor[23,32]. **b** UAG number and editable UAG sites of all genes and essential genes in each chromosome. The editable sites mean the TAGs can be converted to TAAs by cytosine base editors with editing window from position 1-13(base positions are numbered relative to the PAM-distal end of the guide RNA). **c** Kernel density curves for the densities of TAG codons in the GRCh38.p13 build of the human genome obtained using GRIT. The density curves for the chromosomes are normalized to have uniform height and width. Chromosome lengths and total TAG counts are given on the left-hand side.

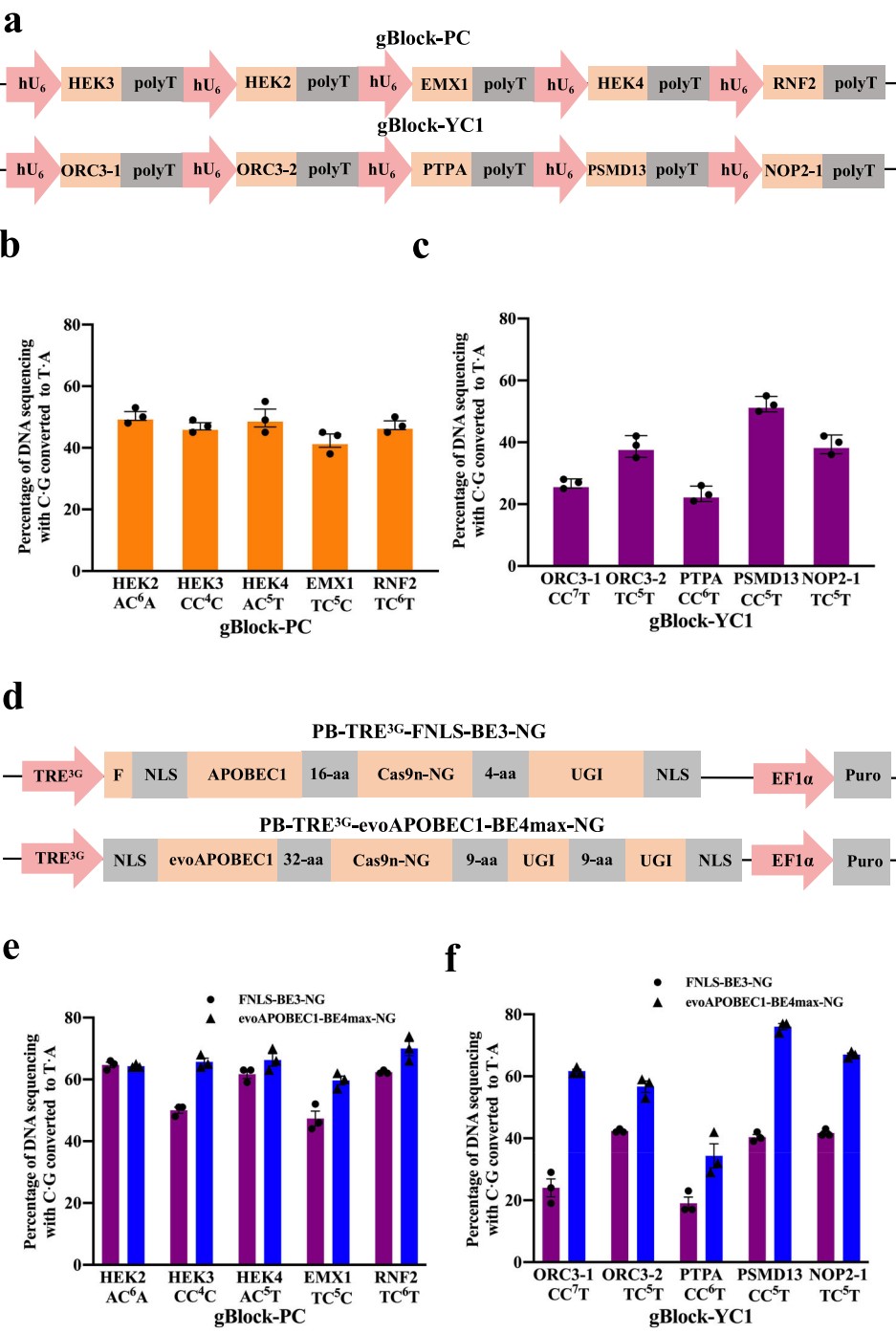

**Fig. 2 Multiplexed base editing in HEK293T cells by gRNAs array. a** Schematic diagram of gBlock-PC and gBlock-YC1. gBlock-PC carries published five sgRNAs targeting at five endogenous loci (HEK2, HEK3, HEK4, EMX1, RNF2) and gBlock-YC1 carries 5 sgRNAs targeting TAG of five genes loci (ORC3-1, ORC3-2, PTPA, PMSD13, NOP2-1). Co-transfected with gblock-PC (**b**) and gBlock-YC1 (**c**) with evoAPOBEC-BE4max-NG into HEK293T cells separately. Frequency (%) of C-to-T conversion was obtained by Sanger sequencing and editR analysis. **d** Schematic diagram of dox-inducible cytidine deaminase piggyBac construct. F Flag tag, NLS Nuclear localization signal, Cas9n-NG, Cas9D10A with recognizing NG PAM. APOBAEC1, rat APOBEC1; evoAPOBAEC1, evolved rat APOBEC1. Frequency (%) of C-to-T conversion in two stable HEK293T cell lines (FNLS-BE3-NG (**e**) and evoAPOBEC1-BE4max-NG (**f**)) transduced with gBlock-PC and gBlock-YC1 separately. In **b**, **c**, **e**, and **f**, dots and triangle represent individual biological replicates ($n = 3$ independent experiments) and bars represent mean values ± s.d.

with piggybacFNLS-BE3-NG[32] and evoAPOBEC1-BE4max-NG respectively (Fig. 2d). We transiently transfected gBlock-PC and gBlock-YC1 separately into each of the two inducible CBE cell lines and the editing efficiency of sgRNAs from the gBlock-PC is ~60–70% across genes in the evoAPOBEC1-BE4max-NG cell line, which is slightly higher than ~45–65% in the FNLS-BE3-NG cell line (Fig. 2e). However, the efficiency of sgRNAs from the gBlock-YC1 is

~30–75% in the evoAPOBEC1-BE4max-NG cell line, which is significantly higher than ~20–40% in the FNLS-BE3-NG cell line (Fig. 2f). Thus, we decided to use the stably expressing evoAPOBEC1-BE4max-NG cell line in subsequent experiments.

Next, we picked 11 single clones from evoAPOBEC1-BE4max-NG stable cell line to transfect with gBlock-YC1. We performed Sanger/EditR and found clone 1 had the highest editing efficiency

at five tested sites compared to the other ten clones (Supplementary Fig. 2a). Western Blot analysis showed that clone 1 had the highest CBE protein expression among all 11 clones, suggesting that high editing efficiency was associated with high CBE protein expression (Supplementary Fig. 2b). Then, we demonstrated that gBlocks were stable during plasmid amplification and transfection into mammalian cells (Supplementary Fig. 3a, b, c), and transfected 10-, 20-, and 30 gBlock pools into evoAPOBEC1-BE4max-NG stable clone 1 to determine the number of gBlock cassettes that can be delivered at one time with good editing efficiency (Supplementary Fig. 4a). We performed Whole Exome Sequencing (WES) and analyzed editing efficiency at all mapping sites separately, and the editing efficiency at each site decreased significantly as targeted sites increased. Moreover, 22 out of 35 mapping sites of the first 52 gene sites (Supplementary Table 1) is highest when 10 gBlocks are delivered compared to 20 and 30 gBlocks (Supplementary Fig. 4b, c and Supplementary Table 2). We also attempted to assemble 10 gRNA cassettes into one vector with DsRed for transfection validation by golden gate cloning, and analyzed clones by SpeI digestion and Sanger sequencing to confirm a successful 43-gRNA array called 43-all-in-one (Supplementary Fig. 5a, b).

**Determination of effective methods for TAG to TAA recoding in single cells using scRNAseq.** To identify an effective strategy for converting TAG to TAA, we applied the following methods followed by mutation detection via single cell RNAseq: (1) Method_1: 10 gBlocks + mCherry-inactivated eGFP reporter[33]; (2) Method_2: 10 gBlocks + mCherry-inactivated eGFP reporter and eGFP cognated sgRNA plasmid[33]; (3) Method_3: 43 sgRNAs all-in-one (Supplementary Fig. 6). We sorted ~1000 single cells from each condition and performed single cell RNA-seq to examine the distribution of each targeting locus across three cell populations (Fig. 1a). Quality control metrics analyses of the samples are shown in Supplementary Fig. 7a, b, c. We mapped a total of 38/52 gene sites, and observed the number of cells decreased as the number of editing sites increased in all three methods and the number of cells with most edited gene sites was the highest in Method_2 (Fig. 3a). We plotted the population density of cells (Fig. 3b) and analyzed editing efficiency of each target and targets with editing events exhibited a bimodal distribution (Fig. 3c and Supplementary Fig. 8). Editing efficiency of each mapped site in each single cell (Fig. 3d) and total editing efficiency of each target in each sample (Fig. 3e) were also analyzed. Collectively, these data show that Method_2 is the most efficient for TAG to TAA replacement.

**Culture and identification of highly modified HEK293T clones.** To further investigate which method generates highly modified expandable clones, we sorted and cultured single cells from populations transfected by Method_2 or Method_3, and got 28/96 and 24/96 single cell clones, respectively. For clones from Method_2, we picked 10 well-edited loci (one from each gBlock based on the previous WES sequencing analysis to validate their delivery), PCR-amplified them, followed by Sanger sequencing and EditR analysis for preliminary screening. The results showed that 4 clones without gBlocks and 24 clones with between 1 to 10 different numbers of gBlocks, and clone 19 contained all 10 gBlocks (Supplementary Fig. 9a). For clones from Method_3, we used 3 out of 10 well-edited loci for screening and found 13 clones had no editing, and 11 clones had 1 to 3 edited sites, of which clone# 11, 20, 21, and 24 had all 3 sites edited (Supplementary Fig. 9b). Then, we performed Sanger sequencing for all targeted loci in 2 highly modified clones (clones 19 and 21). In clone 19 from Method_2, we observed TAG to TAA substitution

at 33/47 genomic sites, of which 9 sites are homozygous, and 14/47 sites are unedited. Clone 21 from Method_3 showed 27/40 desired editing sites, of which 10 are homozygous TAA, and 13/40 sites are unedited (Fig. 4a). This result is consistent with our previous finding detected with scRNAseq. To determine whether editing efficiency could increase with subsequent transfection rounds, we also transfected gBlocks into the highly modified clone 19 using Method_2 and selected clones 19-1, 19-16, and 19-21 from 22/96 clones due to higher editing (Sanger/EditR) in select loci, as compared to the original clone 19.

**Analysis of on- and off-target effects on highly modified HEK293T clones by WGS.** To comprehensively assess on- and off-target efficiencies of CBE genome-wide TAG to TAA conversion, we performed Whole-Genome Sequencing (WGS) at 30X on highly modified clones (19, 21, 19-1, 19-16, 19-21) and the negative control (clone 1, as the mother cell). For on-target editing, the heat map showed 39/47 gene sites have been mapped and 25 to 28 of them are edited in the highly modified clones. Editing efficiency ranges of those editable sites from ~33% to 100%. Clones 19-1, −16, −21 showed improved editing efficiency ranges from ~10% to 40% at several loci compared to clone 19 (Fig. 4b and Supplementary Table 3). This result was consistent with our previous finding detected with Sanger sequencing. To find off-target events, we analyzed the single nucleotide variants (SNVs) and insertion/deletions (Indels) in highly modified clones (19, 21, 19-1, 19-16, 19-21) compared to the control. After subtracting on-targets, SNVs were 23084, 70356, 35700, 42595 and 31530, respectively (Fig. 4c). Further analysis on these clones revealed 277, 805, 419, 470, 358 SNVs, respectively, were located on exons (Fig. 4c), and only 25, 66, 33, 35, 31 SNVs were located in exons of essential genes, respectively (Fig.4d and Supplementary Fig. 10). We classified the SNVs into individual mutation types and found that C·G-to-T·A transitions were the most frequent edits as expected (Fig. 4e), and the number of C·G-to-T·A SNV mutation of clones were 14371, 59464, 25901, 32695, 22080, respectively (Fig. 4f). In addition to SNVs, the number of Indels detected in these clones was 558, 715, 717, 662, 655, respectively, with a small subset located in exons (Fig. 4g) and none in exons of essential genes.

**Evaluation of gene expression and karyotyping of highly modified HEK293T clones.** To examine potential gene expression changes before and after editing in highly modified clones, we performed uniform manifold approximation and projection (UMAP) analysis on the single cell RNA-seq data, and did not observe cell clustering driven by a high number of edits, indicating no significant gene expression change as a result of editing (Supplementary Figs. 11, 12). Next, we analyzed the bulk RNA-seq data for highly modified clones (19, 21 and 11), lowly modified clones (5, 16) and the negative control. We performed on-target analysis, and the results were consistent with those of WGS (Fig. 5a). Gene expression levels in highly modified clones and lowly modified clones were mostly similar in all genes (Fig. 5b–e) and the 43 targeted loci (Fig. 5f). A few genes were differentially expressed between highly, lowly and wild-type negative control clones (Fig. 5b, c and Supplementary Fig. 13a–d), with gene names and gene expression fold change shown in more detail in Fig. 5e, f. We also did GO enrichment analysis of differentially expressed genes (DEG) between the highly modified clones and lowly modified clones, and did not find any gene set enrichment (Supplementary Fig. 13e). So, bulk RNA-seq is an effective method for high-throughput screening of single clones with multi-site editing because it is less costly than WES and WGS, and gene expression changes can also be assessed before and after

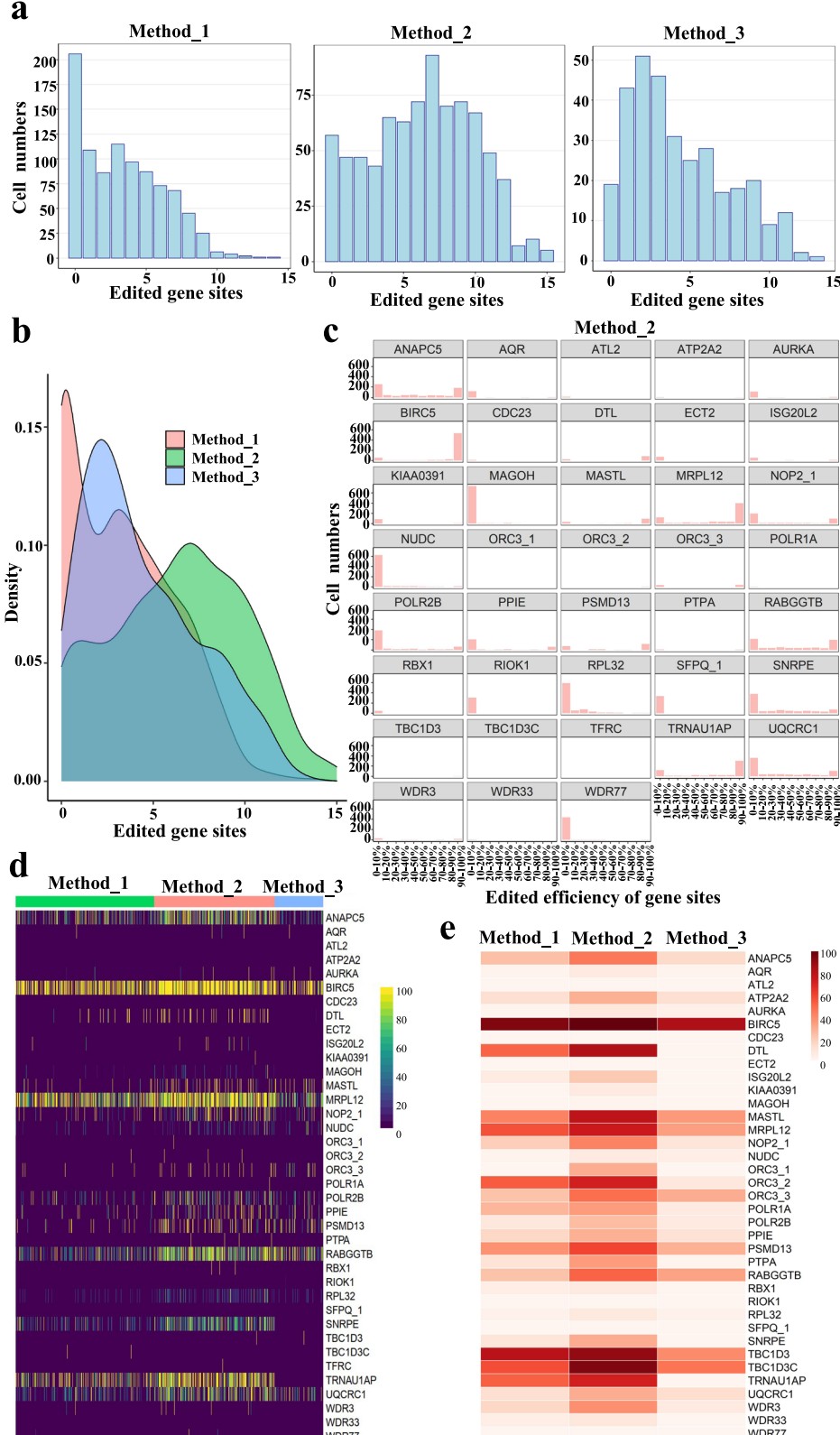

**Fig. 3 Evaluation of experimental strategy for converting TAG to TAA using single cell RNAseq. a** Distribution analysis of cells with different number of modified gene targets in populations with three different delivery methods based on single cell RNAseq. Method_1, delivery 10 gBlocks with mCherry-inactivated eGFP reporter; Method_2, delivery 10 gBlocks with mCherry-inactivated eGFP reporter and eGFP sgRNA plasmids; Method_3, delivery 43-all-in-one with DsRed. **b** Density plot for distribution of number of modified gene targets detected by scRNAseq in 3 populations. **c** For each gene target, distribution analysis of modified cells with different editing efficiency. Counts from method_2 was showed in the plot. **d** Editing efficiency of each sgRNAs in single cells. **e** Heatmap of target "C" editing efficiency in the population with different methods based on converting single-cell RNA-Seq into Bulk RNA-Seq. Editing efficiency was indicated with the intensity of red.

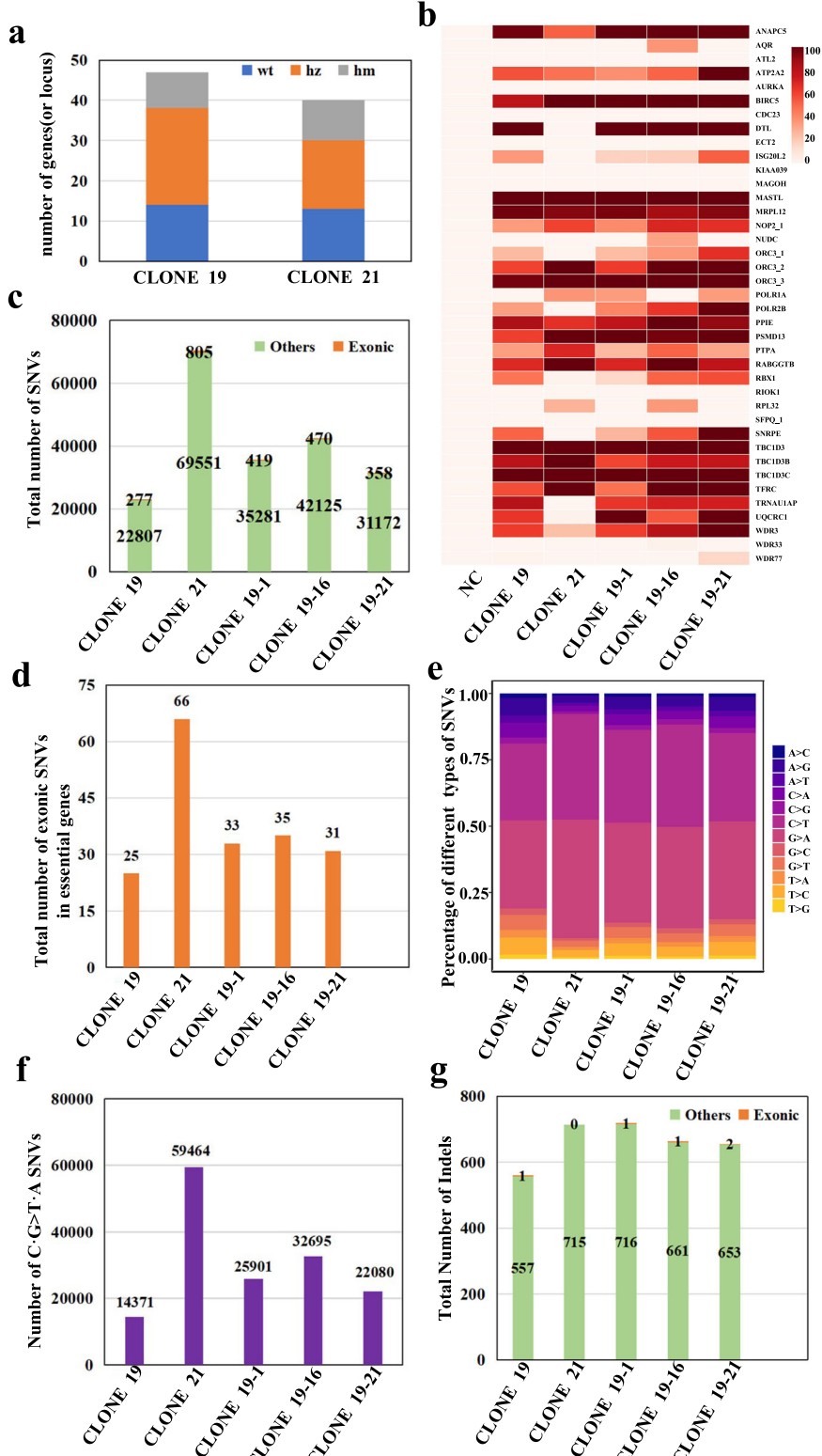

**Fig. 4 Analysis of the genetic changes of highly modified HEK293T clones identified by WGS. a** Allele editing of all target sites in each clone by Sanger sequencing and EditR. wt—no allele editing; hz (heterozygous) – partial allele editing; hm(homozygous) - all allele editing. **b** Heatmap of target "C" editing efficiency for converting TAG to TAA. NC, negative control, HEK293T-BE4max stable cell; clone 19 from method_2, clone 21 from method_3; clone 19-1, 19-16, 19-21 from second transfection by method_2. **c** Number of exonic SNVs (SNVs are located on exons and splicing sites) or other SNVs detected in highly modified clones, as compared to the sequence of the parental HEK293T. The numbers of total SNVs in clone 19, clone 21, clone 19-1, 19-16, 19-21 were 23084, 70356, 35700, 42595 and 31530, respectively. **d** Number of exonic SNVs detected in essential genes. **e** Distribution of different types of SNV changes. **f** Number of detected C·G > T·A SNVs across samples. **g** Total number of exonic indels or other indels detected in highly modified clones. Highly modified clone means this clone has more edited sites than other clones, and editing efficiency of each edited site is above 3%.

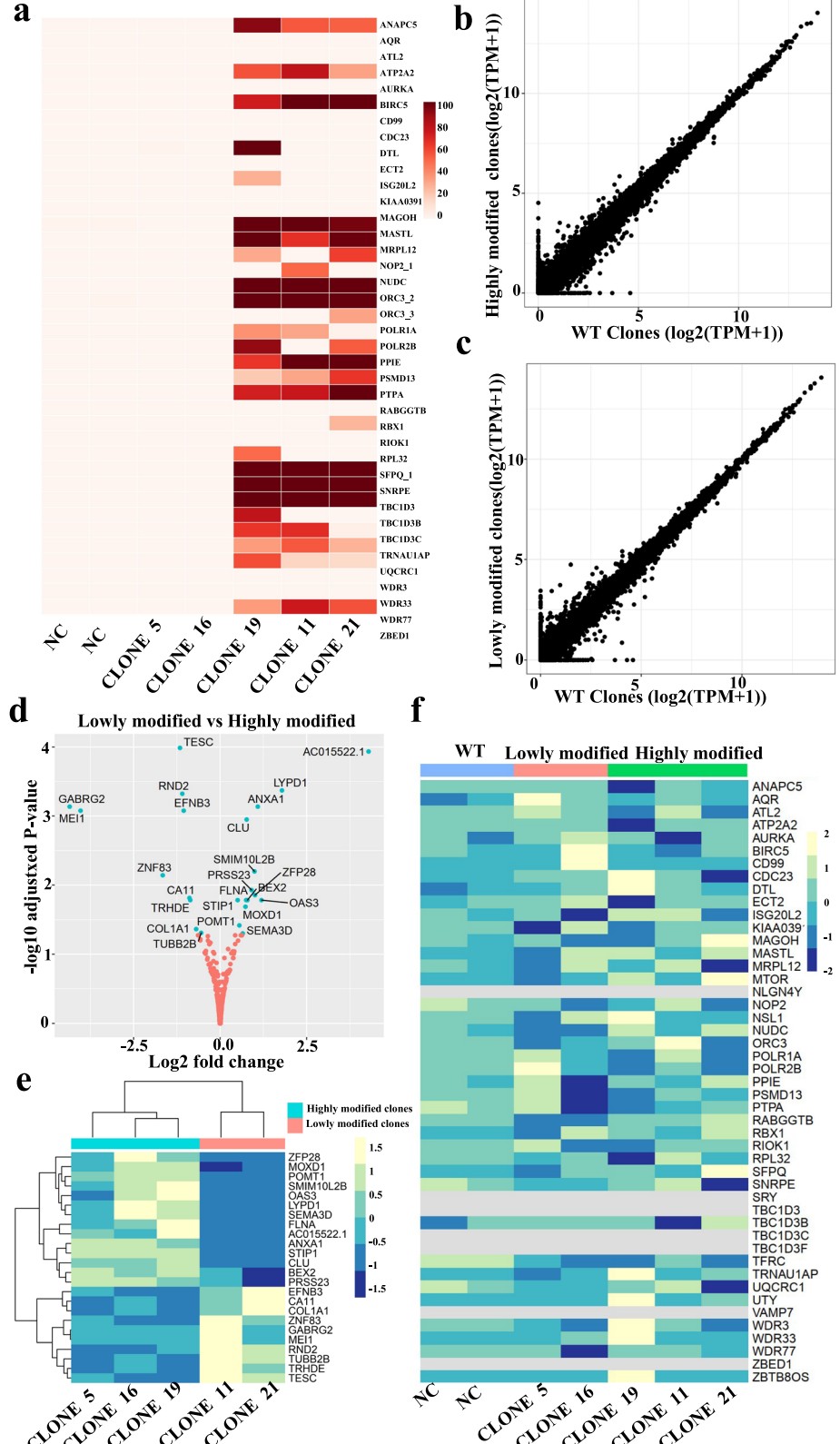

**Fig. 5 On-target and gene expression analysis in highly modified HEK293T clones and lowly modified clones by bulk RNAseq. a** On-target editing efficiency in two negative control (NC) clones, two lowly modified clones (5, 16), and three highly modified clones (19, 21 and 11). **b** Transcriptional correlation of wild-type negative control clones and highly modified clones. **c** Transcriptional correlation of wild-type negative control clones and lowly modified clones. **d** Volcano plot for differentially expressed genes between lowly modified clones and highly modified clones. **e** Heatmap for differentially expressed genes between lowly modified clones and highly modified clones. **f** Expression level of targeted loci in three groups (wild-type negative control clones, lowly and highly modified clones). Highly modified clone means this clone has more edited sites than other clones, and editing efficiency of each edited site is above 3%. The lowly modified clone means the clones have no sites edited after transfection and FACS sorting, wild-type negative control clones were derived from untransfected cells.

editing. We also examined whether unexpected genomic rearrangements had occurred as a result of the multiplexed genome editing. Karyotyping of individual modified clones (Supplementary Fig. 14 and Supplementary Table 4) indicated that there were no observable genomic rearrangements due to multiplex editing.

## Discussion

Here, we firstly reported the GRIT for multiplexed base editing to convert UAG into UAA codons in HEK293T cell line. We determined that of 6700 TAG codons, 6648 are editable across the human haploid genome. More importantly 1947 essential gene TAG codons are in essential genes of which 1937 are editable, based on GRCh38.p13. For TAG codons that cannot be edited by NG-CBE, we plan to edit them by using Cas12a-CBE[34], prime editor[28,35], and HDR[36] in the future. Though GRIT was only tested on human genome data, it could be repurposed for other eukaryotic species for which high-quality genome data similar to GRCh38.p13 is available.

We directly designed and synthesized a gBlock containing five gRNA arrays where editing efficiency is comparable with that of single gRNA array when it is transfected into the stable and doxycycline-inducible HEK293T lines with evoAPOBEC1-BE4max-NG. We also successfully assembled gBlocks into an all-in-one plasmid with 43 gRNAs at one time, which is an array with more sgRNAs than previously reported[37–39] and highly reduced assembly time. Then, we optimized a strategy for TAG to TAA recoding by single-cell RNAseq, which is the first time it has been used to evaluate the outcome of base editor targeting at multiple loci. Using multiplexed base editing, we edited up to 33 of the 47 target sites in a single delivery. Repeating delivery of sgRNAs can increase editing efficiency. This work demonstrates the feasibility of TAG to TAA codons conversion throughout the human genome and establishes a framework for multiplexed genome editing of non-repetitive loci in mammalian cells.

We also observed CBE-mediated off-target burden on the genome, which is consistent with previous studies[40,41]. Yang, Gao et al. reported that BE3, the original CBE, induces random genome-wide mutations at average frequencies of $5 \times 10^{-8}$ per bp and $5.3 \times 10^{-7}$ per bp, when BE3 and single sgRNA was overexpressed in mouse embryos and rice respectively[42]. From a quick calculation for a hypothetical mathematical prediction, in our best clone 19, in order to get 9 homozygous recoded gene termini on target per cell, we also incurred 24 heterozygote off-target mutations in 4764 (predicted) essential genes. To get 1937 precise edits "on-target", we tolerated a Poisson average of 5165 hits (mostly C to T) in off-target 4764 essential genes (including ~1550 homozygous). If we reduce the off-target rate by 60-fold, then we expect <1 homozygous off-target gene per cell, and thus, we predict clones with zero homozygous off-targets. Some genes are more difficult to edit and therefore, CBE-based editing efficiency must significantly improve. Adenine base editors (ABE) have lower off-target burdens, but they are not suitable for recoding TAG. Our group has previously recorded the highest number of CBE-based edits (~6300) in LINE-1 repetitive elements in human cells[43]. However, a single guide RNA was designed for the highest homology among LINE-1 elements, which we predict generates a lower off-target mutation burden compared to multiple gRNAs. In the future, off-target mutations can be ameliorated with less off-target CBE[20,44], RNP[45] and DddA-split base editors[46].

Human recoding is a systematic and complex genome project. Our current study is the initial step of the human genome recoding, which opens the door for subsequent efforts like the *E. coli* recoding and Sc2.0 synthetic genome. Although we can achieve up to 33 genes in single clones via one transfection so far, we can optimize this framework to scale-up to all essential genes or all genes ending with TAG that can be converted to TAA.

Here, we propose several potential strategies: (1) Further developing new base editors with low off-target, high editing efficient and PAM-less/free based on CBE variants[44,47,48], prime editor[28,35] and DdCBE[46,49]; (2) Improve sgRNA delivery capability with a larger gRNA array on BAC or YAC vectors and sgRNA pools; (3) Employ new delivery methods through RNP[50] and synchronous transfection (Supplementary Fig. 15). The next goal in the human recoding roadmap is to utilize all strategies with highly evolved editing tools in a concerted fashion with few to one rounds of edits, assess and optimize recoding efficiencies, followed by massive off-target and bystander mutation cleanup using highly engineered base-editing enzymes.

To make virus-resistant human cells equivalent to *rE.coli*[2,3], we also need to delete the eukaryotic release factor 1 (eRF1), in addition to genome-wide TAG to TAA replacement. However, human cells utilize a single release factor eRF1 (encoded by human *ETF1* gene) which recognizes all three stop codons. The eRF1 cannot be deleted directly, but it has been shown that engineered variants of eRF1 can be made that recognize TAA and TGA with high affinity but TAG with low affinity[51]. Although ectopic expression of selected eRF1 variants E55D from small eRF1 mutant library can increase nonstandard amino acids incorporation via readthrough of amber stop codons, it is no selective readthrough of any stop codons[52]. Thus, a high-throughput comprehensive eRF1 mutagenesis screen is needed for a mutant eRF1 which can replace the endogenous eRF1, and allow normal recognition of UAA and UGA but little or no recognition of UAG as a hypothetical possibility.

In summary, our results represent the first steps to convert TAG to TAA, preliminarily demonstrate the feasibility of TAG to TAA in the human genome, and provide a framework for large-scale engineering of mammalian genomes[53,54]. GRIT can also be developed into a new Computer Aided Design (CAD) platform for writing of larger genomes. Once complete, genome-recoded human cells will offer a unique chassis with extended functionality that could be broadly applicable for biomedicine, especially for making cell therapies or therapeutic production lines resistant to most natural viruses.

## Methods

**Computational design procedure and design rules.** The Genome Recoding Informatics Toolbox (GRIT) provides a Python-based platform for working with human genome data, specifically GRCh38.p13 (GenBank Assembly Accession GCA_000001405.28). The central functions of GRIT are to parse genome data, find TAG codons, and identify guides for base editors with NG protospacer adjacent motifs (PAMs). Using these data, GRIT identifies 6700 total TAG sites (including ones in alternate isoforms of the same genes). Of these, 6648 are editable across the human haploid genome using base editors with editing window from position 1–13. Additionally, 1947 (1937 of which are editable) of the 6700 TAG codons are in genes that do not have strong evidence of nonessentiality. It's noteworthy that this estimate from HGRAss is greater than the number of genes with TAG stop codons in the genome because we consider stop codons in multiple isoforms. It is meant for human genome recoding of TAA to TAG, but it can be used for informatics involving coding DNA sequences, chromosomal sequences, gene essentiality, multiple isoforms, multifunctional sites, base editor guides, and primers. It can easily be run from a desktop computer. Though it was only tested on human genome data, it could be repurposed for other eukaryotic species for which high-quality genome data similar to GRCh38.p13.

The key functions for GRIT are in two python files. The main file, \texttt{GRIT.py} contains sample code and functions to replicate results. There are five functions with docstrings for replicating results in GRIT.py: \texttt{demo}, \texttt{count_total_sites}, \texttt{count_editing_sites}, \texttt{find_genes_to_recode}, and \texttt{get_all_site_data}. Each can be run from the command line. The second file, \texttt{GRIT_utils.py}, contains a \texttt{Chromosome} class, a \texttt{Gene} class, and helper functions. Additionally, \texttt{plot_tag_sites.ipynb} can be run to reproduce Fig. 1c. Inside of \texttt{GRIT_utils.py}, a chromosome object is instantiated. Sites are found that can be directly edited with a C base editor or edited with a "daisy" chain of A and C editors. See the output of \texttt{demo} to see how these are represented. When a chromosome object is instantiated, GRIT will gather data including chromosome name, wildtype sequence, recoded sequence, indices of sites to recode, base editor sites, gene objects, and edit sites that are part of different genes or different codons read in different frames and which two genes they are part of. Gene objects are generally meant to be instantiated automatically and from within the Chromosome class. When

one is instantiated, GRIT gathers data including name, chromosome, strand, wild-type sequence, recoded sequence, active isoform, introns, isoform information, gene essentiality data, and recoding sites.

In both the Chromosome and Gene classes, all methods within them (with the exception of the to_file() method within the Gene class) are *typically* meant to be called only once upon object initialization in order to instantiate class attributes. However, many methods have variable parameters and can be called in an ad hoc manner after initialization with unique arguments to return alternate objects.A Chromosome object is instantiated by passing in a single number or letter giving the chromosome id. When one is instantiated, the __init__() method calls each method inside the class to instantiate the following attributes:

- Chromosome name as a string.
- Full GRCh38.p13 sequence (both wt and recoded versions) as strings.
- A list of two lists giving the indices of all positive strand and negative strand G/C pairs in the chromosome to recode.
- A list of two dictionaries for positive and negative strand daisy-chain sites. Sites are keys, and the values are lists. The first list item is a list of the ABE sites needed if any, and the second is the CBE site. However, if a site is not daisy-chain editable, then the dictionary valye is simply [−1, −1].
- A dictionary of dictionaries. The outer dictionary represents gene names, and the inner dictionaries represent site keys and target sequence values in the same form as above.
- A dictionary with gene name keys and Gene object values for each gene in the chromosome.
- A dictionary with int index keys and list values which contain a pair of strings representing gene names giving edit sites that are part of multiple genes or multiple codons read in different frames and the two genes they affect.

Using a reasonably fast laptop computer (GRIT was largely developed on a Macbook Pro with 16 GB RAM), each chromosome object can be instantiated in well under a minute or two unless the default setting to not create site primers is overridden, in which case it will take much longer.

Gene objects are generally meant to be instantiated from within the chromosome class. When one is instantiated, the __init__() method calls each method inside the class to instantiate the following attributes:

- Gene name as a string.
- Chromosome as the Chromosome object that this gene was instantiated in the __init__() function of.
- Strand as a Boolean with True indicating the positive strand.
- Essentiality information as a list with [0] indicating the number of OGEE-cited essential results and [1] giving the number of noessential ones. If no data is available, this is stored as a string saying "unavailable".
- Isoforms as a list of lists of lists. Each mid-tier list represents a unique isoform, and each innermost list is a pair of start and stop indices for an exon. Isoforms are sorted by CDS length.
- An integer giving the active isoform, meaning that one that is being used for a reference in the rest of the gene class attributes. This will default to iso 0 but can be changed manually or automatically if there is a data error for the first isoform.
- The wildtype and recoded CDS sequences of the active isoform as strings.
- Strings giving the wildtype and recoded "genomic region" beginning at the first start codon for any isoform and ending at the last stop codon including all introns.
- Lists of Cas9 nuclease sites and dictionaries of all base editor sites with sites as keys and gRNA(s) as values.
- Targets, gRNAs, and homology templates (with silent mutations in cas9 sites) for cas9 HR recoding all as lists of ints or strings.
- If a find_all_primers variable at the top of the file is set to True (it defaults False for efficient runtimes), a dictionary giving primer3 results for each site.

The gene class also contains a to_file() method that, but default, is not called upon object instantiation. If called, it will write a.txt file giving all of the gene object's attributes to the working directory.

**Plasmids cloning**. FNLS-BE3-NG was generated using the NEBuilder HIFI DNA Assembly kit (New England Biolabs(NEB) cat# E2621L) according to manufacturer's instructions, by combining a PCR-amplified FNLS-APOBEC1 DNA from pLenti-TRE3G-FNLS-PGK-Puro (Addgene#110847), PCR-amplified Cas9n-NG DNA from pX330-SpCas9-NG (Addgene#117919), PCR-amplified UGI DNA from pLenti-TRE3G-FNLS-PGK-Puro (Addgene#110847) and an NheI/PmeI-digested piggyBac dox-inducible expression vector PB-TRE-Cas9[55] including a puromycin selection marker. The evoAPOBEC1-BE4max-NG DNA from pBT375 (addgene#125616) were cloned between the NotI and PmeI sites of the PB-TRE-Cas9 with NotI restriction enzyme site insertion. NEB Stable Competent E. coli (NEB cat# C3040I) was used following the manufacturer's instructions. Q5 High-Fidelity 2X Master Mix (NEB cat# M0494S) was used for all PCRs. All enzymes and buffers were obtained from New England Biolabs unless otherwise noted. Nuclease-free water (Life Technologies cat# 10977-015) was used for cloning and PCR reactions. All primers and oligos were synthesized by IDT.

**gBlock synthesis and Golden Gate assembly**. All gBlock fragments containing five sgRNA expression cassettes with high fidelity four-base overhang pair[56] after cutting with type IIS restriction enzyme BbsI restriction enzyme were designed and directly sent to be synthesized into PUC57 cloning plasmid by GenScript. Two oligos with BbsI cutting sites were annealed and cloned into a backbone vector with a CMV promoter driving fluorescent protein expression, using SpeI-HF. 10 gBlocks and the backbone plasmid were cutted by BbsI-HF separately, and then gel extraction using gel extraction kit (Zymo Research cat# 11-301 C). gBlock fragments and the backbone plasmid were ligated by T4 DNA ligase (NEB cat# M0202S) at 16 °C overnight. After the ligation reaction, we transformed the 2 µl reaction mix into a competent E. coli strain NEB Stable, according to the manufacturer's protocol. Plasmid DNA from cultures was extracted using a QIAprep spin miniprep kit (cat# 27104) according to the manufacturer's instructions. To validate the multiple sgRNA plasmid, we firstly can roughly check whether the assembly is successful by SpeI cutting. Because There is a SpeI site on either side of the multiple sgRNAs insertion site. When multiple sgRNAs are assembled successfully in the plasmid, two bands will be seen on a gel electrophoresis after the plasmid was cut by SpeI. One band is 4479 bp and another is 22140 bp. Then we verified multiple sgRNAs insertion by Sanger sequencing.

**Cell culture**. HEK293T cells were purchased from ATCC. HEK293T cells were maintained in high-glucose Dulbecco's Modified Eagle Medium (Gibco cat# 11965092) with 10% (v/v) fetal bovine serum (FBS, Gibco cat# 10082147), at 37 °C with 5% $CO_2$ and passaged every 3–4 days, and tested for mycoplasma with Universal Mycoplasma Detection Kit (ATCC® 30-1012 K™) every 4–6 weeks.

**Transient transfection**. Transfection was conducted using Lipofectamine 3000 (Thermo Fisher Scientific cat# L3000015) using the protocol recommended by the manufacturer with slight modifications outlined below. Twenty-four hours before transfection, 50,000 cells were seeded per well in a 48-well plate along with 250 µl of media. For single gBlock and base editor plasmid, a total of 1 ug of DNA (750 ng of base editor plasmid, 250 ng of single gBlock plasmid) and 2 µl of Lipofectamine 3000 were used per well.

**Generation of CBE stable cell lines**. Twenty-four hours before transfection, 500,000 HEK293T cells were seeded per well in a 6-well plate. A total of 4 µg of piggyBac targeting base editor plasmid was transfected with 1 µg of super trans-posase plasmid (SBI System Biosciences cat# PB210PA-1) using Lipofectamine 3000 following the manufacturer's instructions. After 48 h, cells were selected with puromycin (2 ug/ml). Cells were grown for 7–10 days under puro selection, then polyclonal pools used or single-cell sorted by flow cytometry into 96-well plates.

**Detection of gBlocks copy number in HEK 293 T by qPCR**. 1ug gBlock-PC, gBlock-YC1 and gBlock backbone plasmid containing non-repeating sequence insertions of the same size with gBlocks as control were transfected into HEK293T separately. Cells were subjected to standard DNA extraction procedures, and treated with 0.5 U/µl RNaseA (Takara) for 30 min at 37 °C at 12, 24, 48, and 72 h after transfection. qPCR was performed against the housekeeping gene GAPDH using following primers (GAPDH-S: CACCGTCAAGGCTGAGAAC; GAPDH-A: TGGTGAAGACGCCA GTGGA) based on the following protocol[57]. Primers (gBlock-s: GGTGTGAAATA CCGCACAGA and gBlock-a: GGCCGTTACCCAACTTAATCG) for gBlocks were used. Gene Expression Assay (MX3005P; Agilent Stratagene) was used for assaying GAPDH (endogenous control) with the kit (Takara cat#RR420B) in separate reactions. detected with a SYBR Green probe. All reactions were 40 cycles using standard cycling conditions (initial 30 s at 95 °C and 40 cycles of 30 s denaturation at 95 °C and 30 s annealing at 60 °C and extension at 72 °C, finally 1 cycle of 30 s at 95 °C, 30 s at 60 °C, 30 s at 95 °C). The copy number of gBlocks and control plasmid was calculated from their respective CT (cycle threshold) using the linear equation from the respective plasmid standard curve. The plasmid copy number was calculated by dividing the number of plasmids to the number of GAPDH.

**Transfection of gBlock pools and multiple sgRNA plasmids into CBE stable cell**. Twenty-four hours before transfection, 100,000 cells were seeded in each well of 48-well poly-(d-lysine) plates (Corning cat# 354413) along with 300 µl of media with Doxycycline (2 ug/ml), 20 mM cyclic Pifithrin-Alpha (Stem-Cell Technologies cat # 72062) and 20 ng/ml human recombinant bFGF (Stem-Cell Technologies cat# 78003). For the 10 gBlock pool, 200 ng of each gBlock and 3ul of Lipofectamine 3000 were used per well and 20 ng of green fluorescent protein as a transfection control. For the 20 gBlock pool, 150 ng of each gBlock and 3 µl of Lipofectamine 3000 were used per well and 20 ng of green fluorescent protein as a transfection control. For the 30 gBlock pool, 100 ng of each gBlock and 3 µl of Lipofectamine 3000 were used per well and 20 ng of green fluorescent protein as a transfection control. After transfection, Doxycycline was added for another 5 days and then cells were harvested for genomic DNA and editing analysis.

**Single-cell RNAseq**. After transfection, Doxycycline was added for 5 days, then we changed to medium without Doxycycline and continued to culture for 5 days. Single cell isolation was performed by FACS using fluorescence expression. Library preparation for 10X Genomics single-cell RNA sequencing was performed

following manufacturer's instructions for Chromium Next GEM Single Cell 3′ Reagent Kits v3.1. Briefly, after single-cell suspension was acquired from flow sorting, cells and reagents were loaded into Chromium Next GEM Chip G with a targeted recovery of 1000 single cells per sample. Droplet generation, reverse transcription, cDNA amplification, fragmentation and adapter ligation were conducted according to the manufacturer's protocol. Sequencing of the library was performed on the Illumina NovaSeq6000 S1 flow cell (Read 1: 28 cycles, Read 2: 300 cycles, single i7 Index: 8 cycles), with a targeted depth of 300,000 reads per cell.

Raw sequencing reads were processed with Cell Ranger 5.0.0 to generate the gene count matrix. Seurat R package 4.0.1 was used for downstream expression analysis. Due to variance in the sample's sequencing depth, different cell filters were applied. Sample 1 and 2: gene number > 3000, mitochondrial gene percentage < 7; sample 3: gene number > 5000, mitochondrial gene percentage < 10. Normalization was performed using SCTransform function, with the options to regress out variance from mitochondrial gene ration and cell cycle. Principal component analysis was performed with RunPCA function. Top 40 dimensions were used to generate UMAP embedding with the RunUMAP function.

**Single-cell clonal isolation**. After transfection, Doxycycline was added for 5 days and then we changed the medium without Doxycycline to continue to culture for 5 days. Single cell isolation was performed by FACS with fluorescence-based expression, into flat bottom 96-well plates containing 100 μl of DMEM with 10% FBS and 1% Penicillin/ Streptomycin per well. Sorted ellsware incubated for 10–14 days until well-characterized colonies were visible, with periodic media changes performed as necessary. Single cell clones were first dissociated using 20 μl Accutase (STEMCELL Technologies cat# 07920) and enzyme neutralized with 20 μl growth media, and then single cell clones were directly expanded to 24 well plates with 800 μl media for expansion.

**Genomic DNA extraction**. For Sanger sequencing, at 5 days post-transfection, cells were washed with PBS, lysed in 200 μl of QuickExtract™ DNA Extraction Solution (Lucigen Cat. # QE09050) per well of 48-well plates, and genomic DNA (gDNA) was extracted using the manufacturer's protocol. Briefly, the sorted plates were sealed, vortexed and heated at 65 °C for 6 min then 98 °C for 2 min. All primers for Sanger sequencing are shown in supplementary Table 5. For whole-exome and whole-genome sequencing, DNA was extracted using the PureLink™ Genomic Plant DNA Purification Kit (Thermo Fisher cat# K183001) according to the manufacturer's protocol.

**Western blotting**. For Western blot, HEK293T clone cells were lysed 5 days after Doxycycline was added using RIPA buffer supplemented with proteinase and phosphatase inhibitors. Total protein was quantified using the BCA kit (Beyotime cat# P0012). 20 μg per well of total protein was separated by electrophoresis using a 15-well 4–12% Tris-Gly and transferred to a PVDF membrane at 300 mA before blocking with 10% skimmed milk powder for 2 h at 4 °C. PVDF membranes were incubated with a 1:1000 dilution of anti-GAPDH (ABclonal, A19056) and a 1:1000 dilution of anti-Cas9 (ABclonal, A14997) overnight. Then, membranes were incubated with a 1:1000 dilution of HRP Goat Anti-Rabbit IgG (H + L) (ABclonal, AS014) for 2 h and visualized using Tanon imager (Supplementary Fig. 4b).

**Whole-exome sequencing and whole-genome sequencing**. For whole exome sequencing, 1.5-5ug DNA processed with Exome Kit Agilent SureSelect XT Human All Exon V5, and sequenced with Illumina NovaSeq6000 S4 (2 × 150 bp) at 50X coverage. Processing, sequencing and preliminary analysis conducted by Psomagen (South Korea). For whole-genome sequencing, library generation and sequencing were carried out using the Illumina Truseq Kit with 30X coverage at Beijing Genomics Institute (BGI) Hong Kong.

**Preparation of RNA libraries for bulk RNA sequencing**. HEK293T cells cultured in 6-well plates were washed with PBS and harvested by adding 600 μl TRIzol (Life Technologies cat#15596026) directly to the cells. Total RNA was extracted with Direct-zol RNA Miniprep Plus kit (Zymo Research cat# R2070) following manufacturer's protocol. RNA integrity was confirmed with the presence of 18 S and 28 S bands on a 2% E-Gel EX (Invitrogen cat# G402002). RNA libraries were prepared with NEBNext Poly(A) mRNA Magnetic Isolation Module (NEB cat# E7490L) and NEBNext Ultra II Directional RNA Library Prep Kit for Illumina (NEB cat# E7760L). 500 ng RNA was used at input, and the quality of final libraries were confirmed by qPCR and TapeStation. Sequencing of the libraries was performed by the Biopolymers Facility at Harvard Medical School using NovaSeq6000.

**On-target analysis for whole exome sequencing, bulk RNA sequencing and whole-genome sequencing**. BWA was used to map sequencing reads to the reference human genome (hg38). Bam files were further analyzed with CRISPResso2 version 2.0.31[58]. An input gene list with 20 bp gRNA sequence (no PAM included) for each target and chromosome coordinates for a 41 bp mapping region with the target edits centered were generated. CRISPRessoWGS mode was applied to detect genomic changes in 52 selected regions among samples with customized usages -wc

−15 -w 10 -p 5 and base edit-related usages -base_-edit --conversion_nuc_from C (--conversion_nuc_from G) --base_editor_output. "SAMPLES_QUANTIFICA-TION_SUMMARY.txt" was used to quantify the percentage of modified reads and "Selected_nucleotide_percentage_table_around_sgRNA.txt" was used to quantify the desired base edits (C > T or G > A) for each target. Subsequently, heat maps for percentage of modified reads and desired base edits were plotted respectively.

**Off-target analysis by whole-genome sequencing**. We called SNPs and indels using somatic tumor-normal approach (using a control sample as a normal, and edited samples as 'tumor'), and two variant callers (mutect2 followed by FilterMutectCalls (from gatk package v4.2.0.0) and strelka2 v2.9.10) were applied and only variants passed filters were selected. For mutect2-called variants, reference counts and alternative counts were calculated based on tier 1 A/C/G/T counts with those for strelka2-called variants were pre-calculated. Shared variants from vcf files were selected by bedtools v2.29.2 to confirm a variant to be called (a similar approach was taken by Zuo et al.[59]). The SNVs and indels were separated based on the length of the reference and alternative allele. Annovar[60] was used to further annotate the SNVs and indels using refGene, a gene-based annotation, to illustrate the distribution of different variant types. Whether detected variants were in essential genes were also examined.

**On-target analysis for scRNAseq**. BAM files were generated from fastq files using Cell Ranger 5.0.0. BAM files were filtered for cell barcodes passed quality control and variants were called using CRISPResso2 as described in "On-target analysis for Whole exome sequencing, bulk RNA sequencing and whole-genome sequencing". Edited targets were defined as targets with mapped and at least 2 reads of desired C or G. Individual cells with different numbers of edited targets were quantified and plotted to demonstrate the distribution among different delivery and enrichment methods as shown in Fig. 3a–c and overlapped density plot was shown in Fig. 3d. For each target, on-target editing efficiency was also plotted as shown in Fig. 3g.

**Evaluate gene expression levels by RNA sequencing data analysis**. STAR 2.5.2b was used for alignment of reads and quantification of gene expression. Briefly, a human genome reference index was built using genome primary assembly (ftp://ftp.ebi.ac.uk/pub/databases/gencode/Gencode_human/release_27/GRCh38. primary_assembly.genome.fa.gz) and annotation file (ftp://ftp.ebi.ac.uk/pub/ databases/gencode/Gencode_human/release_27/gencode.v27.primary_assembly. annotation.gtf.gz) from GENCODE. Per gene counts were generated using STAR -quantMode GeneCounts. Differential gene expression analysis was performed using DESeq2 with raw counts from STAR. Genes with an adjusted $p$-value < 0.05 were called differentially expressed. For figures that used transcripts per million (TPM) values, TPM counts were generated using Salmon. TPM for each gene was produced by aggregating the TPM value from all transcripts from the same gene.

**Karyotype analysis of highly modified single cell clones**. Highly modified HEK293T clones (clone 19, clone 21) were expanded and karyotypically compared with the control groups and the wild-type HEK293T. Actively growing cells were passaged 1–2 days prior to sending to BWH CytoGenomics Core Laboratory. The cells were received by the core at 60–80% confluency. Chromosomal count, variances and abnormalities were investigated.

**Statistics and reproducibility**. All statistical analyses were performed on at least three biologically independent experiments using GraphPad prism9. Detailed information on exact sample sizes and experimental replicates can be found in the individual figure legends. Tests for statistically significant differences between groups were performed using a two-tailed Student's $t$-test and all $P < 0.05$ were considered significant.

**Reporting summary**. Further information on research design is available in the Nature Research Reporting Summary linked to this article.

# Data availability
Sequencing data have been deposited in the NCBI Sequence Read Archive database with accession code PRJNA730314. All plasmids in this study will be available upon reasonable request. Source data are provided with this paper.

# Code availability
Codes have been uploaded to the Github repository github.com/thestephencasper/GRIT, including code and files for reproducibility.

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

## Acknowledgements

This work (G.C., E.H., and Y.C.) was supported by a pilot Harvard Quadrangle Fund for Advancing Seeding Translational Research, Aging and Longevity-Related Research Fund at Harvard Medical School, and U.S Department of Energy grant DE-FG02-02ER63445. Y.C. was supported by National Natural Science Foundation of China (No. 32101173). C.L. was supported by Strategic Priority Research Program of Chinese Academy of Sciences (Nos. XDPB18, XDB29050501), National Natural Science Foundation of China (Nos. 32025022, 3201101136), Shenzhen Grants (Nos. KQTD2015033117210153). We thank GenScript for gblock DNA synthesis and Chun-Ting Wu for helpful discussions.

## Author contributions

Y.C., E.H., S.C., S.L. and K.Y. performed experiments. Y.C., G.C, E.H., A.C., S.L., and C.L. designed the experiments and analyzed the data. Y.C., E.H., A.C., S.L., C.L. and G.C wrote the paper with input from all authors.

## Competing interests

All G.M.C COI are listed here: http://arep.med.harvard.edu/gmc/tech.html. Yuting Chen has submitted patent applications (nos. 202111153709.1, 202111151234.2, PCT/CN2021/121730 and PCT/CN2021/121750) based on the results reported in this study. All other authors declare no competing financial interests.
