## [Peer Review File · Nature Communications]

Reviewers' Comments:

Reviewer #1:

Remarks to the Author:

The manuscript "Multiplex base editing to convert TAG into TAA codons in the human genome" by Chen et al describes a method to simultaneously introduce C-to-T mutations in multiple TAG codons in HEK cells using evoBE4max. The use of scRNAseq to measure co-occurring editing in single cell levels demonstrates an intriguing approach to quantify multiplexing outcomes. The authors also performed very comprehensive characterizations of on-target edits, off-target edits, karyotyping, and gene-expression levels. In general, it is interesting that this paper showed multiplex base editing at up to 39 genes in single clones. However, I am not convinced that the goal that the authors set out in the introduction, to recode all TAG codons to TAA for a genome-wide scale of 6648 editable genes, can be achieved by using the method described in this paper, and it's unclear to me what the significance is of the progress that was made in the current work. Some specific comments are as follow.

1. Figure 1 is misleading and does not represent the actual experimental workflow in this paper. Only two transfection rounds had been performed, and the second round was performed after the selection and clonal expansion of the highly edited cells from the first round.
2. It is not clear what exactly the 10-, 20-, and 30 gBlocks are. It was shown that a gBlock contains 5 different gRNA cassettes. Do gBlocks in each pool all contain different gRNA cassettes, meaning 10-, 20- and 30 gBlock pools were targeting 50, 100 and 150 different genes respectively? If it is true, it doesn't really make sense to compare the editing efficiencies of the first 52 gene sites among these 3 strategies. The transfections of 20- and 30 gBlocks would inherently afford lower efficiencies than that of 10 gBlocks because the CBE protein was "occupied" editing at more genes. In addition, the use of the 43-all-in-one construct is confusing to me given that this construct doesn't even contain all the 52 genes the authors used for characterization. I am also worried that not all gRNAs were expressed properly from this construct because of its highly repetitive sequence. The experiment would be more credible if the authors verified the expressions of gRNAs – maybe using a Northern blot?
3. The description for the method_1 and method_2 is confusing. (What is an eGFP cognated sgRNA plasmid?) It would be helpful if there is a schematic explanation. If I understand correctly based on the referenced literature, I am confused by the rationale behind comparing these 3 methods. Method_2, which uses an CBE activity reporter, is always going to outperform method_1, which only uses a transfection reporter, and method_3, which does not even have a reporter. If the authors aim to compare different gRNA delivery strategies, I suggest keeping the report system constant.
4. In the analysis of on- and off- target section, the authors' claims could be more convincing if they show the numerical data in the main text instead of having readers to identify the color scales in the figures. For example, what percentage would be defined as "highly edited", and how much were the editing percentages increased after the second-round transfection? What is the frequency of C>T SNVs across the whole genome?
5. The term "SNV mutation ratios" is not defined and it was not clear what control they were normalized to. The authors claimed that both SNV and indel ratios (both around 0.3, which is incredibly high unless it's a percentage, not a ratio) were "very low" in all samples but did not provide any reasons or evidence from previous studies to support such a claim.
6. The differential gene expressions are interesting. The paper would benefit if the authors would elaborate more on the implications of the genes that are being differentially expressed. Are they related in any way? Was a gene set enrichment analysis performed?
7. The discussion section is highly speculative. I am not convinced by the quick calculation on the off-target, because the second-round of transfection can increase the number of off-target by up to almost two-fold. (Clone 19-16: 42595 vs clone 19: 23084). How many rounds of transfections are needed to get homozygous editing of 1937 genes given that one round only affords 9? Considering the time and cost, it seems like it would be unrealistic to achieve such a goal based on the method developed in this paper?
8. Supplemental Figure 1 appears to be missing part b

Reviewer #2:

Remarks to the Author:

This manuscript describes the nascent steps aimed at recoding the human genome, which is an ambitious goal. The authors first present a computational tool (GRIT) used to parse the human genome, identify all TAG codons, and guide their conversion to a synonymous codon, TAA. The authors use this computational program to guide a targeted set of experiments that use C base editors (CBE) to make the TAG-TAA modification. The authors then describe a series of experiments primarily aimed at advancing the multiplex capability of CBEs through generating arrays on a set of plasmid constructs or stable cell lines and fluorescent reporter readouts, RNA-seq and exome sequencing to measure editing efficiencies. The authors ultimately identify a specific method that allowed them to introduce 33 edits simultaneously. This is probably the most notable technological advance of the study. The authors conclude by conducting a number of off-target assays and functional characterizations to measure unintended mutations and any pleiotropic effects of their recoding. They identified significant frequency of off-targets, but did not detect any aberrant phenotypic consequences yet. Should the authors employ such a strategy for recoding the entire genome, the absolute number of off-targets would be vast and something that the authors should project and address in this study.

Overall, this study presents some of the first steps of an effort that seeks to perform whole-genome recoding of human cells. This is based on some of the Church lab's prior work on recoding bacteria. I think this manuscript is interesting and warrants publication in Nature Communications. I also believe that there are some shortcomings in the paper that need to be addressed prior to publication:

- The writing throughout the manuscript can be improved for clarity and word usage. Below I provide a number of specific instances that require attention, but the same level of scrutiny needs to be applied throughout the manuscript.
- The abstract and introduction need to be revised to more accurately describe the current study and cite the appropriate literature when describing relevant precedence. In addition to the specific examples below, I also think the authors should cite a number of papers over the past ~decade from the SC 2.0 consortium who are redesigning the yeast genome. Part of their designs involve UAG to UAA recoding and the key papers here should be cited (eg, PMIDs: 21918511, 28280199, 24674868). Another recoding project should also be cited: PMID: 28499033
- The figures throughout the paper are hard to read – both because images are blurry and font sizes are very small. This made reviewing data throughout the ms very challenging and, in some cases, not possible. As such, the reader/reviewer cannot fully review and interpret the data and understand conclusions from this study. The authors need to correct this throughout the ms.
- The last paragraph of the introduction provides five key points that explains their choice of recoding TAG to TAA. I think the manuscript would benefit by expanding upon this discussion to more fully explain the rationale and approach.
- Related to the prior point, I also found that some important design considerations to recoding the TAG stop codon were missing. For example, prior work in E coli demonstrated that TAG to TAA recoding enabled the deletion of one of the bacterial release factors. Since human cells only have a single release factor, how would an equivalent step be implemented in human cells? The authors need to explain their strategy for addressing this problem.

Specific Points to address:

1. Abstract: I suggest a number of re-phrasing to more accurately convey background and this ms:
 - a. "Large-scale recoding" to "Whole-genome recoding"
 - b. "novel amino acids" to more conventional usage: "noncanonical amino acids" or "nonstandard amino acids"
 - c. Here, we extend this to human cells...As written, this sentence gives the impression that the current ms extends recoding, novel amino acids, biocontainment and virus resistance to human cells. This is not the case and should be revised to more accurately convey the initial steps described in this ms toward that goal
2. Introduction, line 46 – "disabling tRNA" is only relevant for sense codons, but stop codons require deleting release factors. In the case of many eukaryotes, like human, only one release factor is present. I don't believe the authors address how this would be addressed and this is

important. Presumably the release factor is essential for peptidyl cleavage. Thus, what do the authors propose to do once all TAG codons have been converted to TAA? Since the RF can't be deleted, I would assume it needs to be mutated, which could present a challenge. This should be added for completeness.

3. Introduction, lines 47-48: This should be stated as a hypothesis as not enough characterization has been done in the field of recoding to absolutely state that the proteome will be the same.

4. Introduction, lines 48: "recoding confers virus resistance" – two key papers that empirically provide this are missing: PMIDs: 27426981 & 30375330

5. Introduction, line 49: the repurposing of blank codons to incorporate nonstandard amino acids has a solid citation from 2001, which is old and not relevant to the repurposing of blank codons. The first recoded cell was not demonstrated for another ~10 years. I suggest the authors replace this citation with their Reference #2

6. Introduction, line 50: another key citation is missing on biocontainment: PMID: 25607356

7. Introduction, lines 56-58: in addition to Ref #2, two additional papers should be cited that demonstrate virus resistance as mentioned in point #4 above

8. Introduction, lines 66-72: present key rationale for the factors that guided recoding of TAG. I recommend breaking each point into separate sentences expanding upon the rationale with citations for each of the five points. From my perspective, the most compelling reasons are the first two – but, it would be more insightful to delineate the rationale in more depth.

9. Lines 75-91: introduce GRIT – the computational algorithm to automate the design of recoding. This is a valuable part of this effort, but very little information is described about how the software works. Is this software available online or via Github? I also recommend a more detailed description should be included in the text and/or supplement.

10. Figure 1: general comments: this figure uses very small fonts and images that are hard to resolve. Additionally, parts of the figure are blurry. I recommend improving aesthetic for clarity. a: I don't see how this part of the figure describes GRIT? It illustrates the design of a gRNA pool in an array that is transfected into HEK293 cells followed by FACS and other types of characterizations. Building off the prior point, more description is required and a dedicated figure for GRIT is warranted. This sub-figure illustrates a different workflow. Part b: how are the authors defining a base editable site? Part c: what biological insights are the authors trying to convey here?

11. Line 95: typo: "exiting" should be "existing"

12. Lines 98-115 describe the construction of different CBE cassettes and cell lines. Again, more information is needed to understand and appreciate the differences in design and corresponding results. Related to this, a specific question that I think the authors should address is how much sequence similarity exists across the constructs housing 5 sgRNAs? Expanding upon this, do the authors observe any genetic instability? Could that explain some of the decrease in transfection efficiency from the prior papers cited? Also, why did the authors seek to construct stable vs. transient cell lines? Upon considering their goals of whole-genome UAG-UAA editing, that approach seems limited and laborious.

13. Supplementary Fig 2a: It is my understanding that the data represents the frequency of edits/site across the population? It would be informative to understand the distribution of # edits/cell or clone.

14. Supplementary Fig 3 shows important data comparing the deliver of 10, 20, or 30 g-blocks, but this figure is not resolvable. The fonts are tiny and the different conditions are illustrated in an easily digestible way. I recommend that the authors improve the clarity and representation of this figure.

15. Supplementary Fig 4: another interesting experiment with a large (43) array of sgRNAs. I'll pose a similar question as above: did the authors study the stability of these plasmids over culture? What is the amount of sequence redundancy? What mutations explain the constructs that were not the correct size.

16. Lines 134-136 reference a single-cell RNA-seq experiment and Fig 1a, which was referenced for the GRIT software previously. This is confusing and should be corrected

17. Lines 130-144 describes another important experiment that tests different approaches at multiplexing but not explained clearly. Also Fig 3 also suffers from possessing tiny plots and unreadable fonts so nearly impossible to interpret the data. Please increase font size for clarity.

18. Lines 146-163 describes another important experiment to quantify multiplex editing efficiency. The approach is unclear to me – how do the authors know which "10 well-edited loci" to pick a priori? I believe these data are plotted in Supp Fig 6 – given the importance of such data, I recommend that the authors consider relocating to a main figure.

19. Lines 164-184 describes important experiments that study off-target frequencies. The associated Fig 4 suffers from the same issues as prior figures – data and fonts are tiny and blurry and therefore hard to read and interpret. Increasing size and clarity is critical for the reader to interpret the data and the authors' interpretation.
20. Lines 185-202 present informative characterization experiments that warrant more detailed description and interpretation. Also Fig 5 had small fonts and was hard to resolve. More specific referencing of the sub-parts would be informative in understanding the conclusions that the authors are drawing from the data. Lastly, Fig 5f is hard to interpret – what is meant by "high-edit", "low_edit", vs. "wild_type"?
21. I appreciate the karyotyping – I would suspect only very gross chromosomal arrangements could be detected here – what type of resolution or noticeable result would you expect from such an assay?
22. Line 205: indicates 6700 TAG codons, which is inconsistent with earlier in the text: 6888 (line 83) – this discrepancy needs to be reconciled.
23. Lines 237-238: Although this study takes the first steps toward recoding a human genome, certainly a daunting and challenging task, I would hesitate to state that it "demonstrates feasibility". I think this study takes the first steps toward such a goal.
24. Lines 239-240: More information in this paper should be included to describe GRIT in detail and made accessible to reviewers and readers.
25. Lines 240-243: would the authors describe this work as "genetically modified human cells", which is quite generic. Their goals are more focused on whole-genome recoding.

Dear reviewers,

We appreciate your invaluable comments. We thought deeply about the insightfulness of your comments, and we have made substantial revisions to incorporate them into our manuscript.

Firstly, we would like to emphasize that the objective of our study is the first attempt made to date to provide the framework of converting TAGs to TAAs across the human genome. At first, we developed a software - GRIT used to identify all TAG codons in the human genome, and designed gRNAs to guide their conversion to a synonymous codon, TAA. Though the software was only tested on human genome data, it could be repurposed for other eukaryotic species with high quality reference genomes that are fully annotated. Then, we developed efficient TAG to TAA conversion methods by multiple gRNA arrays delivery with CBE edited reporter that allows for enriching highly edited cells. We evaluated the outcome of base editing in a single cell by scRNA-seq for populations, and bulk RNAseq and WGS for modified single clones.

We acknowledge that our current study is the initial step of the human genome recoding, which opens the door for subsequent efforts like the *E. coli* recoding and Sc2.0 synthetic genome. Although we can achieve up to 33 genes in single clones via one transfection so far, we propose potential strategies to scale-up to all essential genes ending with TAG that can be converted to TAA with a larger gRNA array on BAC or YAC vectors, less off-target CBE (PMIDs:33247095, 32345976) and several transfection rounds.

Below find a point-by-point response to your comments:

Reviewer #1 (Remarks to the Author):

The manuscript “Multiplex base editing to convert TAG into TAA codons in the

human genome” by Chen et al describes a method to simultaneously introduce C-to-T mutations in multiple TAG codons in HEK cells using evoBE4max. The use of scRNAseq to measure co-occurring editing in single cell levels demonstrates an intriguing approach to quantify multiplexing outcomes. The authors also performed very comprehensive characterizations of on-target edits, off-target edits, karyotyping, and gene-expression levels. In general, it is interesting that this paper showed multiplex base editing at up to 39 genes in single clones. However, I am not convinced that the goal that the authors set out in the introduction, to recode all TAG codons to TAA for a genome-wide scale of 6648 editable genes, can be achieved by using the method described in this paper, and it’s unclear to me what the significance is of the progress that was made in the current work. Some specific comments are as follow.

1. Figure 1 is misleading and does not represent the actual experimental workflow in this paper. Only two transfection rounds had been performed, and the second round was performed after the selection and clonal expansion of the highly edited cells from the first round.

Thank you for your comments. We have redesigned Figure 1a. and added n=2 in the Figure 1 legend.

2. It is not clear what exactly the 10-, 20-, and 30 gBlocks are. It was shown that a gBlock contains 5 different gRNA cassettes. Do gBlocks in each pool all contain different gRNA cassettes, meaning 10-, 20- and 30 gBlock pools were targeting 50, 100 and 150 different genes respectively? If it is true, it doesn’t really make sense to compare the editing efficiencies of the first 52 gene sites among these 3 strategies. The transfections of 20- and 30 gBlocks would inherently afford lower efficiencies than that of 10 gBlocks because the CBE protein was “occupied” editing at more genes. In addition, the use of the 43-all-in-one construct is confusing to me given that this construct doesn’t even contain all the 52 genes the authors used for characterization. I am also worried that not all gRNAs were expressed properly from

this construct because of its highly repetitive sequence. The experiment would be more credible if the authors verified the expressions of gRNAs – maybe using a Northern blot?

Thank you for your comments. We have added supplementary tables showing 10-, 20-, and 30 gBlocks containing sgRNAs sequence in the supplementary Excel data of the revised manuscript. 20 gBlock contains the first 10 and 30 gBlock contains the first 20, so 10-, 20- and 30 gBlock pools have in common 50 gRNAs (labeled with yellow color). Although 10-, 20- and 30 gBlock pools were targeting 50, 100 and 150 genes sites respectively, we add doxycycline into the medium for 24 h to induce expression of CBE protein before transfection, and continue adding doxycycline for 5 days after transfection. Thus, we believe that CBE protein is sufficient for all 3 groups, and think these three groups are comparable.

We apologize for the confusion about the 43-all-in-one construct. We attempted to assemble 10 gBlocks containing 50gRNAs targeting 52 genes sites, but we were only able to verify a 43-all-in-one construct via Sanger sequencing. Although this construct doesn't contain all 50 sgRNAs, we also believe this construct can be used as a stand-in to compare bigger gRNA arrays vs. individual gRNAs.

We agree that the repetitive sequence maybe affect gene expression in mammalian cells. However, several previous studies have reported that they have successfully generated 10-30 functional gRNA array all-in-one plasmid (PMIDs: 27178736, 28849913 30939239), which is a similar plasmid structure with our 43 all -in-one and multiple gRNAs can be expressed from 43 all-in-one plasmid by placing each under the control of an individual Pol III U6 promoters and terminators. Although we did not verify the expression of all 43 gRNAs by northern blot, our data showed that each sgRNA from 5-gRNA array is functional. We affirm this as the genomic sites of single clone 21 were targeted by most of sgRNA in the 43-all-in-one construct (method_3). Unedited sites from 43-all-in-one construct use are same with clone 19 from where the cells were transfected by 10 gBlocks (method_2) (Figure 4b). We believe this demonstrates poorly edited sites, rather than sgRNA expression disparities.

3. The description for the method_1 and method_2 is confusing. (What is an eGFP cognated sgRNA plasmid?) It would be helpful if there is a schematic explanation. If I understand correctly based on the referenced literature, I am confused by the rationale behind comparing these 3 methods. Method_2, which uses an CBE activity reporter, is always going to outperform method_1, which only uses a transfection reporter, and method_3, which does not even have a reporter. If the authors aim to compare different gRNA delivery strategies, I suggest keeping the report system constant.

We apologize for the confusion and thank you for your suggestion. We have made a schematic diagram for method_1, method_2 and method_3 in Figure S5 in the revised manuscript. In method_3, 43-all-in-one construct contained a CMV promoter driving a Dsred fluorescent reporter (Figure S4, S5), as a transfection reporter which is consistent with method_1 and method_2 where we used mCherry-inactivated eGFP reporter. We chose the backbone of 43-all-in-one because it's convenient for a golden gate assembly. Here we are trying to compare which of these approaches is more effective in implementing the TAG-to-TAA transition. Method_1 means 5-gRNA array pool with transfection reporter, method_2 means 5-gRNA array pool with CBE editing reporter, method_3 means 43-gRNA array (bigger array) with transfection reporter. We believe this setup comparing these three groups is reasonable and rational.

4. In the analysis of on- and off- target section, the authors' claims could be more convincing if they show the numerical data in the main text instead of having readers to identify the color scales in the figures. For example, what percentage would be defined as "highly edited", and how much were the editing percentages increased after the second-round transfection? What is the frequency of C>T SNVs across the whole genome?

Thank you for your comments. We have added numerical data description in the main text. Highly modified clone means this clone has more editing sites than other clones and editing efficiency of each editing sites is above 3%, which have also been added to the Figure 4 legend. The increased editing percentage is on average ~10-40% at the

edited sites after the second-round transfection, and the editing efficiency of sites of each clone are also shown in supplementary table 2. We have described the total number of C·G>T·A SNV in main text of revised manuscript and included them in Figure 4f.

“For on-target editing, the heat map showed 39/47 gene sites have been mapped and 25 to 28 of them are edited in the highly modified clones. Editing efficiency ranges of those editable sites from ~33% to 100%. Clones 19-1, -16, -21 showed improved editing efficiency ranges from ~10% to 40% at several loci compared to clone 19 (Fig. 4b and Supplementary Table 2).”

5. The term “SNV mutation ratios” is not defined and it was not clear what control they were normalized to. The authors claimed that both SNV and indel ratios (both around 0.3, which is incredibly high unless it’s a percentage, not a ratio) were “very low” in all samples but did not provide any reasons or evidence from previous studies to support such a claim.

Thank you for your comments. we also think the SNV mutation ratios description is not rigorous. Thus, we have removed the description about SNV mutation ratios and added number of C·G to T·A SNV in Figure 4f in the revised description in the main text of the revised manuscript. Yang, Gao et al. reported that when overexpressed in mouse embryos and rice, BE3, the original CBE, induces random genome-wide mutations at average frequencies of 5×10^{-8} per bp and 5.3×10^{-7} per bp, respectively when they use single sgRNA (PMIDs: 30819928, 30819931 and 32042165). We can’t get the exact calculation of this number because HEK293T hypotetraploid.

“After subtracting on-targets, SNVs were 23084, 70356, 35700, 42595 and 31530, respectively (Fig. 4c). Further analysis on these clones revealed 277, 805, 419, 470, 358 SNVs, respectively, were located on exons (Fig 4c), and only 25, 66, 33, 35, 31 SNVs, respectively, were located in exons of essential genes (Fig.4d and Supplementary Fig. 7). We classified the SNVs into individual mutation types and found that C·G-to-T·A transitions were the most frequent edits as expected (Fig. 4e), and the number of C·G-to-T·A SNV mutation of clones were 14371, 59464, 25901, 32695,

22080, respectively (Fig. 4f). In addition to SNVs, the number of Indels detected in these clones was 558, 715, 717, 662, 655, respectively, with a small subset located in exons (Fig. 4g) and none in exons of essential genes.”

6. The differential gene expressions are interesting. The paper would benefit if the authors would elaborate more on the implications of the genes that are being differentially expressed. Are they related in any way? Was a gene set enrichment analysis performed?

We agree that differential gene expressions are very interesting; however only few genes are differentially expressed as shown in Figure 5d, e and Supplemental Figure 12a, b, c, d. We also did GO analysis and could not find any functional enrichment (Supplemental Figure 12e).

7. The discussion section is highly speculative. I am not convinced by the quick calculation on the off-target, because the second-round of transfection can increase the number of off-target by up to almost two-fold. (Clone 19-16: 42595 vs clone 19: 23084). How many rounds of transfections are needed to get homozygous editing of 1937 genes given that one round only affords 9? Considering the time and cost, it seems like it would be unrealistic to achieve such a goal based on the method developed in this paper?

Thank you for your comments. We have revised this part in the discussion about quick calculation. Here the quick calculation is a hypothetical mathematical prediction discussion. We cannot give the exact model of off-target of multiplex base editing base on 2 rounds of transfections. Although we do not know the exact number of transfection rounds to achieve the goal now, we propose potential strategies to scale-up for all essential genes ending with TAG to be converted to TAA with a larger BAC- or YAC-based gRNA array, CBE with reduced off-target effects (PMIDs:33247095, 32345976) and several transfection rounds using our framework.

8. Supplemental Figure 1 appears to be missing part b

Thank you very much for the careful review. We have fixed it in the revised

manuscript.

Reviewer #2 (Remarks to the Author):

This manuscript describes the nascent steps aimed at recoding the human genome, which is an ambitious goal. The authors first present a computational tool (GRIT) used to parse the human genome, identify all TAG codons, and guide their conversion to a synonymous codon, TAA. The authors use this computational program to guide a targeted set of experiments that use C base editors (CBE) to make the TAG-TAA modification. The authors then describe a series of experiments primarily aimed at advancing the multiplex capability of CBEs through generating arrays on a set of plasmid constructs or stable cell lines and fluorescent reporter readouts, RNA-seq and exome sequencing to measure editing efficiencies. The authors ultimately identify a specific method that allowed them to introduce 33 edits simultaneously. This is probably the most notable technological advance of the study. The authors conclude by conducting a number of off-target assays and functional characterizations to measure unintended mutations and any pleiotropic effects of their recoding. They identified significant frequency of off-targets, but did not detect any aberrant phenotypic consequences yet. Should the authors employ such a strategy for recoding the entire genome, the absolute number of off-targets would be vast and something that the authors should project and address in this study.

Overall, this study presents some of the first steps of an effort that seeks to perform whole-genome recoding of human cells. This is based on some of the Church lab's prior work on recoding bacteria. I think this manuscript is interesting and warrants publication in Nature Communications. I also believe that there are some shortcomings in the paper that need to be addressed prior to publication:

- The writing throughout the manuscript can be improved for clarity and word usage. Below I provide a number of specific instances that require attention, but the same

level of scrutiny needs to be applied throughout the manuscript. (Fixed)

- The abstract and introduction need to be revised to more accurately describe the current study and cite the appropriate literature when describing relevant precedence. In addition to the specific examples below, I also think the authors should cite a number of papers over the past ~decade from the SC 2.0 consortium who are redesigning the yeast genome. Part of their designs involve UAG to UAA recoding and the key papers here should be cited (eg, PMIDs: 21918511, 28280199, 24674868).

Another recoding project should also be cited: PMID: 28499033. (Fixed)

- The figures throughout the paper are hard to read – both because images are blurry and font sizes are very small. This made reviewing data throughout the ms very challenging and, in some cases, not possible. As such, the reader/reviewer cannot fully review and interpret the data and understand conclusions from this study. The authors need to correct this throughout the ms. (Fixed)

Thank you for your comments and suggestion. We have carefully checked the whole manuscript and corrected all the words for clarity in the revised manuscript. We have updated the manuscript by incorporating your suggestion and remaking the new figures in the revised manuscript.

- The last paragraph of the introduction provides five key points that explains their choice of recoding TAG to TAA. I think the manuscript would benefit by expanding upon this discussion to more fully explain the rationale and approach.

Thank you for your comments. We have expanded this section in the revised manuscript.

“We selected amber stop code TAG for the following reasons: 1) Previously published papers reported that recoded E. coli showing nonstandard amino acids incorporation and multiple viruses resistance^{2,3}; 2) TAG is the least commonly used codon in the human genome that allows for fewer edits; 3) TAG could be theoretically edited to TAA using C to T base editors (CBE)¹⁹, and increase flexibility in gRNAs design as TAG denotes the end of the gene, thus reducing concern for CBEs-induced bystander

edits²⁰ effects on gene transcription and translation . ”

• Related to the prior point, I also found that some important design considerations to recoding the TAG stop codon were missing. For example, prior work in E coli demonstrated that TAG to TAA recoding enabled the deletion of one of the bacterial release factors. Since human cells only have a single release factor, how would an equivalent step be implemented in human cells? The authors need to explain their strategy for addressing this problem.

This is an excellent point. We elaborated that in the Discussion Section of our revised manuscript.

“To make viruses-resistant human cells like rE.coli(PMID: 24136966,34083482), we also need to delete the eukaryotic release factor 1 (eRF1), besides genome-wide TAG to TAA replacement. However, human cells utilize a single release factor eRF1 (encoded by human ETF1gene) which recognizes all three stop codons. The eRF1 cannot be deleted directly, but it has been shown that engineered variants of eRF1 can be made that recognize TAA and TGA with high affinity but TAG with low affinity⁴². Although ectopic expression of selected eRF1 variants E55D from small eRF1 mutant library can increase nonstandard amino acids incorporation via readthrough of amber stop codons, it is no selective readthrough of any stop codons⁴³. Thus, a high-throughput comprehensive eRF1 mutagenesis screen is needed for a mutant eRF1 which can replace the endogenous eRF1, and allow normal recognition of UAA and UGA but little or no recognition of UAG as a hypothetical possibility.”

Specific Points to address:

1. Abstract: I suggest a number of re-phrasing to more accurately convey background and this ms:
 - a. “Large-scale recoding” to “Whole-genome recoding” (Fixed)
 - b. “novel amino acids” to more conventional usage: “noncanonical amino acids” or “nonstandard amino acids” (Fixed)
 - c. Here, we extend this to human cells...As written, this sentence gives the impression

that the current ms extends recoding, novel amino acids, biocontainment and virus resistance to human cells. This is not the case and should be revised to more accurately convey the initial steps described in this ms toward that goal (Fixed)

Thank you for your suggestion. We have revised the manuscript by incorporating this suggestion.

“Whole-genome recoding has been shown to enable nonstandard amino acids, biocontainment and viral resistance in bacteria. Here we take the first steps to extend this to human cells demonstrating exceptional base editing to convert TAG to TAA for 33 essential genes via a single transfection, and examine base-editing genome-wide (observing ~ 40 C-to-T off-target events in essential gene exons). We also introduce GRIT, a computational tool for recoding. This demonstrates the feasibility of recoding, and highly multiplex editing in mammalian cells.”

2. Introduction, line 46 – “disabling tRNA” is only relevant for sense codons, but stop codons require deleting release factors. In the case of many eukaryotes, like human, only one release factor is present. I don’t believe the authors address how this would be addressed and this is important. Presumably the release factor is essential for peptidyl cleavage. Thus, what do the authors propose to do once all TAG codons have been converted to TAA? Since the RF can’t be deleted, I would assume it needs to be mutated, which could present a challenge. This should be added for completeness.

Thank you for your comments. We have revised this description in the revised manuscript.

“Recoding is a promising application of genome engineering which involves replacing all instances of a particular codon in a genome with synonymous codons, and deleting the corresponding transfer RNA (tRNA) or release factor 1 (RF1) in prokaryotes while replacing endogenous eukaryotic release factor 1 (eRF1) by engineered eRF1 variants as a hypothetical possibility in eukaryotes.”

Regarding release factors, we elaborate in the Discussion Section of our revised manuscript.

*“To make viruses-resistant human cells like *rE.coli*(PMID:*

24136966,34083482), we also need to delete the eukaryotic release factor 1 (eRF1), besides genome-wide TAG to TAA replacement. However, human cells utilize a single release factor eRF1 (encoded by human ETF1gene) which recognizes all three stop codons. The eRF1 cannot be deleted directly, but it has been shown that engineered variants of eRF1 can be made that recognize TAA and TGA with high affinity but TAG with low affinity⁴². Although ectopic expression of selected eRF1 variants E55D from small eRF1 mutant library can increase nonstandard amino acids incorporation via readthrough of amber stop codons, it is no selective readthrough of any stop codons⁴³. Thus, a high-throughput comprehensive eRF1 mutagenesis screen is needed for a mutant eRF1 which can replace the endogenous eRF1, and allow normal recognition of UAA and UGA but little or no recognition of UAG as a hypothetical possibility.”

3. Introduction, lines 47-48: This should be stated as a hypothesis as not enough characterization has been done in the field of recoding to absolutely state that the proteome will be the same.

Thank you for your comments. We have removed this sentence in the revised manuscript.

4. Introduction, lines 48: “recoding confers virus resistance” – two key papers that empirically provide this are missing: PMIDs: 27426981 & 30375330

Thank you for your suggestion. We have cited them in the revised manuscript.

5. Introduction, line 49: the repurposing of blank codons to incorporate nonstandard amino acids has a solid citation from 2001, which is old and not relevant to the repurposing of blank codons. The first recoded cell was not demonstrated for another ~10 years. I suggest the authors replace this citation with their Reference #2

Thank you for your suggestion. We have fixed in the revised manuscript.

6. Introduction, line 50: another key citation is missing on biocontainment: PMID: 25607356

Thank you for your suggestion. We have fixed in the revised manuscript.

7. Introduction, lines 56-58: in addition to Ref #2, two additional papers should be cited that demonstrate virus resistance as mentioned in point #4 above

Thank you for your suggestion. We have fixed in the revised manuscript.

8. Introduction, lines 66-72: present key rationale for the factors that guided recoding of TAG. I recommend breaking each point into separate sentences expanding upon the rationale with citations for each of the five points. From my perspective, the most compelling reasons are the first two – but, it would be more insightful to delineate the rationale in more depth.

Thank you for your suggestion. We have fixed it in the revised manuscript.

“We selected amber stop code TAG for the following reasons: 1) Previously published papers reported that recoded E. coli showing nonstandard amino acids incorporation and multiple viruses resistance^{2, 3}; 2) TAG is the least commonly used codon in the human genome that allows for fewer edits; 3) TAG could be theoretically edited to TAA using C to T base editors (CBE)¹⁹, and increase flexibility in gRNAs design as TAG denotes the end of the gene, thus reducing concern for CBEs-induced bystander edits²⁰ effects on gene transcription and translation . ”

9. Lines 75-91: introduce GRIT – the computational algorithm to automate the design of recoding. This is a valuable part of this effort, but very little information is described about how the software works. Is this software available online or via Github? I also recommend a more detailed description should be included in the text and/or supplement.

Thank you for your comments. We have added the description in the main text of the revised manuscript. This software is available via Github (Codes have been uploaded to the Github repository github.com/thestephencasper/GRIT, including code and files for reproducibility). We have also added more description in the method section of supplementary material in the revised manuscript.

“The key functions for GRIT are in two python files. The main file, `GRIT.py` contains sample code and functions to replicate results. There are five functions with docstrings provided for replicating results in `GRIT.py`: `demo`, `count_total_sites`, `count_editing_sites`, `find_genes_to_recode`, and `get_all_site_data`. Each can be run from the command line. The second file, `GRIT_utils.py`, contains a `Chromosome` class, a `Gene` class, and helper functions. Additionally, `plot_tag_sites.ipynb` can be run to reproduce Figure 1c.

Inside of `GRIT_utils.py`, a chromosome object is instantiated. Sites are found that can be directly edited with a C base editor or edited with a “daisy” chain of A and C editors. See the output of `demo` to see how these are represented. When a chromosome object is instantiated, GRIT will gather data including chromosome name, wildtype sequence, recoded sequence, indices of sites to recode, base editor sites, gene objects, and edit sites that are part of different genes or different codons read in different frames and which two genes they are part of. Gene objects are generally meant to be instantiated automatically and from within the `Chromosome` class. When one is instantiated, GRIT gathers data including name, chromosome, strand, wildtype sequence, recoded sequence, active isoform, introns, isoform information, gene essentiality data, and recoding sites.”

10. Figure 1: general comments: this figure uses very small fonts and images that are hard to resolve. Additionally, parts of the figure are blurry. I recommend improving aesthetic for clarity. a: I don't see how this part of the figure describes GRIT? It illustrates the design of a gRNA pool in an array that is transfected into HEK293 cells followed by FACS and other types of characterizations. Building off the prior point, more description is required and a dedicated figure for GRIT is warranted. This sub-figure illustrates a different workflow. Part b: how are the authors defining a base editable site? Part c: what biological insights are the authors trying to convey here?

Thank you for your comments. We have redesigned and improved the quality of Figure 1 in the revised manuscript. Part a: We have added description about GRIT. Part b: The editable sites mean the TAGs can be converted to TAAs by cytosine base editors with editing window from position 1-13 (base positions are numbered relative to the PAM-distal end of the guide RNA) in Figure 1 Legend. Part c: This figure shows the densities of TAGs across chromosomes and the image of TAG distribution in each chromosome.

11. Line 95: typo: “exiting” should be “existing” (Fixed)

Thank you very much for the careful review. We have fixed it in the revised manuscript.

12. Lines 98-115 describe the construction of different CBE cassettes and cell lines. Again, more information is needed to understand and appreciate the differences in design and corresponding results. Related to this, a specific question that I think the authors should address is how much sequence similarity exists across the constructs housing 5 sgRNAs? Expanding upon this, do the authors observe any genetic instability? Could that explain some of the decrease in transfection efficiency from the prior papers cited? Also, why did the authors seek to construct stable vs. transient cell lines? Upon considering their goals of whole-genome UAG-UAA editing, that approach seems limited and laborious.

Thank you for your comments. We have shown the diagram of two CBEs and gBlocks in Figure 2. These two CBE lines were generated based on published papers (PMIDs: 29969439, 31332326) separately. We have added the sequence of gBlock-PC in supplementary material, with the same sequence identified by the same color. We conducted dozens of rounds of transformations of 5-sgRNA array plasmids, then we measured the concentration and sequenced them. We did not observe any mutations at insertion sequences. Thus, the constructs with 5 sgRNAs should be overall genetically stable

We observed that the editing efficiency of each gRNA transfection using a 5-gRNA

array was lower than that of a single gRNA. The reason for the low editing efficiency of sgRNA array may be that targeting 5 sites simultaneously is more cytotoxic than targeting only one site.

The reason for generating a CBE stable line is to help us deliver only sgRNA array plasmids at the time of the multiplex editing experiments, to control the integrated CBE copy number in clonal populations, and to remove CBE from the genome by Excision-only PiggyBac Transposase.

13. Supplementary Fig 2a: It is my understanding that the data represents the frequency of edits/site across the population? It would be informative to understand the distribution of # edits/cell or clone.

Thank you for your comments. What we show in Supplementary Fig 2a is the editing efficiency at single CBE clones. We derived 11 single clones from the population of drug-resistant CBE stable cell lines and used gBlock-YC1 to test the editing efficiency at each clone.

14. Supplementary Fig 3 shows important data comparing the delivery of 10, 20, or 30 g-blocks, but this figure is not resolvable. The fonts are tiny and the different conditions are illustrated in an easily digestible way. I recommend that the authors improve the clarity and representation of this figure.

Thank you for your suggestion. we have fixed it in the revised manuscript.

15. Supplementary Fig 4: another interesting experiment with a large (43) array of sgRNAs. I'll pose a similar question as above: did the authors study the stability of these plasmids over culture? What is the amount of sequence redundancy? What mutations explain the constructs that were not the correct size.

Thank you for your comments. That's very interesting question. We conducted dozens of rounds of transformations of the large (43) array plasmid, and then we measured its concentration and sequenced them, and we did not observe any mutation at insertion sequences after the array was cloned. Editing efficiency after repeated transfections

was similar to the editing efficiency of the same sites after the first transfection experiment with the large (43) array plasmid. Our large array should be overall genetically stable. We also have added the sequence of a large (43) array in supplementary material. The redundancy of the array rests on using the same hU6 promoter driving the expression of each gRNA, and the same gRNA scaffold that is repeated for each gRNA used in the array. This repetitive structure could have contributed to the assembly of 43-, not 50-gRNA construct.

16. Lines 134-136 reference a single-cell RNA-seq experiment and Fig 1a, which was referenced for the GRIT software previously. This is confusing and should be corrected

We apologize for the confusion. We have redesigned Figure 1a to show the GRIT software and the single-cell RNA-seq experiment.

17. Lines 130-144 describes another important experiment that tests different approaches at multiplexing but not explained clearly. Also Fig 3 also suffers from possessing tiny plots and unreadable fonts so nearly impossible to interpret the data. Please increase font size for clarity.

Thank you for your comments. We have added supplementary figure 5 to show a schematic explanation of these three different approaches. We have also remade Figure 3.

18. Lines 146-163 describes another important experiment to quantify multiplex editing efficiency. The approach is unclear to me – how do the authors know which “10 well-edited loci” to pick a priori? I believe these data are plotted in Supp Fig 6 – given the importance of such data, I recommend that the authors consider relocating to a main figure.

Thank you for your comments and suggestions. These 10 well-edited loci were selected based on WES data analysis of delivery of 10-, 20-, or 30 g-blocks (Supplementary Figure 3). We also have incorporated Supplementary Fig 6c in main

Figure 4a.

19. Lines 164-184 describes important experiments that study off-target frequencies. The associated Fig 4 suffers from the same issues as prior figures – data and fonts are tiny and blurry and therefore hard to read and interpret. Increasing size and clarity is critical for the reader to interpret the data and the authors' interpretation.

Thank you for your comments. We have remade Figure 4.

20. Lines 185-202 present informative characterization experiments that warrant more detailed description and interpretation. Also Fig 5 had small fonts and was hard to resolve. More specific referencing of the sub-parts would be informative in understanding the conclusions that the authors are drawing from the data. Lastly, Fig 5f is hard to interpret – what is meant by “high-edit”, “low_edit”, vs. “wild_type”?

Thank you for your comments. We have remade Figure 5 in the revised manuscript.

The “high_edit” clone means this clone has more editing sites than other clones and editing efficiency of each editing sites is above 3% after transfection of gRNA array and FACS sorting. The “low_edit” clone means the clones have no sites edited after transfection and FACS sorting, “wild_type” clones were derived from untransfected cells. We have also added this description in the figure 5 legend.

21. I appreciate the karyotyping – I would suspect only very gross chromosomal arrangements could be detected here – what type of resolution or noticeable result would you expect from such an assay?

Thank you for your comments. We performed a classic karyotype analysis of the “high edit clones”, “low edit clones” and WT clones, all of which were normal demonstrating that genome recoding can be conducted without gross chromosomal abnormalities.

22. Line 205: indicates 6700 TAG codons, which is inconsistent with earlier in the text: 6888 (line 83) – this discrepancy needs to be reconciled.

Thank you for your comments. We revised this point in our revised manuscript. The total number of TAG codons is 6700.

23. Lines 237-238: Although this study takes the first steps toward recoding a human genome, certainly a daunting and challenging task, I would hesitate to state that it “demonstrates feasibility”. I think this study takes the first steps toward such a goal.

Thank you for your comments. We have revised our manuscript to reflect this point.

“In summary, our results represent the first steps to convert TAG to TAA, preliminarily demonstrate the feasibility of TAG to TAA in the human genome, and provide a framework for large-scale engineering of mammalian genomes.”

24. Lines 239-240: More information in this paper should be included to describe GRIT in detail and made accessible to reviewers and readers.

Thank you for your comments. We have added description about GRIT in the main text and supplementary Method Section of the revised manuscript. Please refer to the answer to question 9.

25. Lines 240-243: would the authors describe this work as “genetically modified human cells”, which is quite generic. Their goals are more focused on whole-genome recoding.

Thank you for your comments. We have rewritten this sentence in the revised manuscript.

“Once complete, genome-recoded human cells will offer a unique chassis with extended functionality that could be broadly applicable for biomedicine, especially for making cell therapies or therapeutic production lines resistant to most natural viruses.”

Reviewers' Comments:

Reviewer #1:

Remarks to the Author:

The authors have responded to most points adequately, and the manuscript has improved significantly on clarity and data presentations. But I think there are still several improvements could be made and a few extra concerns that need to be addressed prior to publication.

1. The figures are much better after the revision. Yet, some are still blurry or misaligned.

Specifically:

- Lots of the graphs, for example figure 1b, figure 2b/c/e/f, figure 3a and figure 4d, are still blurry (the plot, data labels, axis labels, legends) but their axis titles are clear. I assume the authors generated the plots in one software but add the axis titles in another. This problem can be easily solved by exporting the images as vector graphics or just with higher PPI. If it is possible, it would be better to generate all the chart elements in one software so that the font styles and sizes are consistent and coordinated.
- The boxes in figure 2a/d are not aligned and vary in sizes. Also, some annotations, e.g., PSMD13 and NOP2-1, go outside of their boxes. I would suggest adjusting the font sizes to improve the aesthetic.

2. Line 88-106: I agree with reviewer #2 that GRIT is one of the highlights of this manuscript and should be described in more detail. However, I think the readability of this part in the main text would benefit from having more explanations on the design procedure and rules (like those in the supplementary method) instead of getting too bogged down in the details of each python file. Also, the Github link doesn't seem to be working when I try to visit.

3. Line 147-155: Related to my previous comment #2. Thank you for clarifying the gRNA composition in the gBlock pools and the 43-all-in-one construct. To me, this part is very interesting because the authors here demonstrated multiplexed base editing in human cells at a number of different sites that exceeds any previous studies to my knowledge.^{1,2} As the authors mention in the rebuttal, by taking advantages of a stable and inducible CBE cell line, this system overcomes the Cas9 "bottlenecking" problem. I agree with the authors that, within this setting, it is valid to compare gRNA delivery strategies with increasing number of guides, but I don't think the editing efficiencies of a small collection of 22 common sites, while ignoring 28, 78, and 128 sites respectively, can reflect the true performance of each strategy. With the WES data, the authors have the capability to analyze all the sites and use statistics such as mean, median or total editing efficiencies to represent the performance of each strategy and conduct statistical tests for the comparisons among them. I am curious whether the 20, 30 gBlock pools could have better editing in those non-overlapping sites or could achieve higher total C-to-T conversions. In any case, I presume this analysis would be very informative on determining the theoretical gRNA "ceiling" in HEK cells, and very valuable to the field of base editing.

4. Line 157-161: Related to my comment #3 in the first revision. I appreciate the schematics; it is much clearer now. I would agree with the authors that under such a setup, method_2 is the best. Meanwhile, I notice that the authors put much emphasis on the 43-all-in-one construct as a highlight of this paper. But just like I mentioned previously, it is unfortunate that there is no head-to-head comparison between the 10 gBlock pool strategy and the 43-all-in-one construct, because the method_2 and method_3 use different reporter systems. Given that an activity reporter usually outperforms a transfection reporter, it is still unclear whether or not the 10 gBlock pool strategy is actually better than the 43-all-in-one construct.

5. Line 196-197: Related to off-target analysis. I am not sure if clone 1 is the right control for the off-target analysis because it didn't go through the same clonal expansion process as those highly modified clones, especially 19-1, -16 and -21 which were expanded twice from clone 1. It is well documented that clonal expansion leads to genetic diversification.^{3,4} Zuo et al. found 94% of the CBE off-targets are C-to-T mutations, whereas, from a quick calculation, only ~60-80% from this analysis.⁵ Is the high portion of C-to-non-T off targets be a concern of this method? If a better control is not available, I would recommend addressing this issue in the main text.

6. Line 260-267: Related to comment #7 in my previous review. I still think this calculation is not

rigorous and too speculative. It seems to oversimplify the task without properly addressing the daunting challenges of recoding the whole genome. In both the rebuttal and the discussion, the authors stated that the off-target problem can be mitigated by using CBE with reduced off targets. However, those CBEs come with reduced on-target activity as a trade-off -- they won't be as efficient as evo-BE4max used in this paper and it means more rounds of transfection may be needed. Given only 9 homozygous mutations are obtained by using the most efficient CBE available, just recoding 1937 essential genes entails hundreds of rounds of transfection, certainly not "several" rounds as the authors state. In addition, on-target C-to-non-T editing (mostly C-to-G) is never mentioned in this manuscript, which would change a TAG to a TAC encoding tyrosine and result in continued and inappropriate translation of the mRNA into the 3'UTR region. Would the chance of having non-stop mutations be a concern of this method? (BE-HIVE <https://www.crisprbehive.design/> can potentially predict C-to-G editing)

7. All in all, I think this paper has already showed impressive data on multiplexed base editing and demonstrated its potential to be further expanded to a larger scale. Thus, in the discussion session, I would recommend spending more efforts to address the roadblocks and challenges so that people in the field can improve upon them.

References:

1. McCarty, N. S., Graham, A. E., Studená, L. & Ledesma-Amaro, R. Multiplexed CRISPR technologies for gene editing and transcriptional regulation. *Nat. Commun.* 11, 1281 (2020).
2. Webber, B. R. et al. Highly efficient multiplex human T cell engineering without double-strand breaks using Cas9 base editors. *Nat. Commun.* 10, 5222 (2019).
3. Greaves, M. & Maley, C. C. Clonal evolution in cancer. *Nature* 481, 306–313 (2012).
4. Kakiuchi, N. & Ogawa, S. Clonal expansion in non-cancer tissues. *Nat. Rev. Cancer* 21, 239–256 (2021).
5. Zuo, E. et al. Cytosine base editor generates substantial off-target single-nucleotide variants in mouse embryos. *Science* 364, 289–292 (2019).

Reviewer #2:

Remarks to the Author:

In many instances, the authors responded to the reviewers' comments and have improved the manuscript. This study does present some interesting new multiplex genome editing technology in human cells for the very early steps toward recoding the human genome. These advances warrant publication. That said, there are several parts of the manuscript that can still be strengthened. For example, the language in certain parts of the text can be clarified and improved for specificity and accuracy. Also, some figures still appear blurred or distorted and need to be revised before any publication. Some additional comments to address:

- Introduction, paragraph 1: the newly revised second sentence is a run-on sentence and should be broken down into multiple sentences that first describe prior work on recoding in bacteria (with citations) followed by how this blueprint can be applied to human recoding and eRF1 engineering. You may want to first describe prior recoding work, as done in the subsequent sentences and paragraphs of the intro, and then transition to how this can be applied to recoding human in the last paragraph of the intro
- Results section on GRIT software. I appreciate the authors incorporating more information about this software into the results, but I recommend that the text in the results section describe the design, operations and key results of the software, not the detailed code, which is currently present. Such information should be moved to the methods.
- How many instances of alternate isoforms of TAG sites were found in the human genome? This would be interesting information to include.
- Of the 6700 TAG sites, why are 52 of those non-editable by base editors? What do the authors propose as the plan to edit those sites?
- I cannot find a reference to Fig 1a. I also encourage the authors to enhance visualization for the final publication.
- Fig 1: b: This figure is still distorted and blurred and needs to be corrected before publication; c:

- Fig 2: b, e, and f: title contains non-English characters and the x-axis is cut-off; a & d: lots of different colors and hard to read the font describing each genetic component. Please clarify.
- The authors response to both Reviewers' questions regarding the stability of their 30 gblocks does not fully address the question and demonstrate stability. The authors should consider performing experiments that directly investigate this question – both for future work in this project and for others in the field who wish to replicate the multiplexing experiment described in this study. I also think their concluding remarks in response to this question read in a speculative manner.

Dear reviewers,

We appreciate your invaluable comments again. We have made substantial revisions to incorporate them into our manuscript.

We would like to emphasize that the objective of our study is the first attempt made to date to provide the framework of converting TAGs to TAAs across the human genome. Firstly, we developed a software - GRIT- to parse genome data and identify all TAG codons in the human genome, and designed gRNAs for base editors to guide their conversion to a synonymous codon, TAA. Though the software was only tested on human genome data, it could be repurposed for other eukaryotic species with high quality reference genomes that are fully annotated. Then, we developed efficient TAG to TAA conversion methods by multiple gRNA arrays delivery with CBE edited reporter that allows for enriching highly edited cells. We also analyzed the outcome of multiplexed base editing in single cell from population by scRNA-seq. Finally, we evaluated the highly modified single clones by WGS, bulk RNAseq and karyotyping.

Human recoding is a systematic and complex genome project. Our current study is the initial step of the human genome recoding, which opens the door for subsequent efforts like the *E. coli* recoding and Sc2.0 synthetic genome. Although we can achieve up to 33 genes in single clones via one transfection so far, we can optimize this framework to scale-up to all essential genes or all genes ending with TAG that can be converted to TAA. In the discussion section of the revised manuscript, we propose several potential strategies: 1) developing new base editors with low off-target and high editing efficient based on CBE variants (PMIDs:33247095,32345976,32424272), prime editor (PMIDs:31634902,34653350) and DdCBE (PMIDs: 32641830,35379961); 2) improving sgRNA delivery capability with a larger gRNA array on BAC or YAC vectors and sgRNAs pool; 3) trying new delivery methods through RNP (PMID:28585549) and synchronous transfection (Supplementary Fig.15). The next goal in the human recoding roadmap is to utilize all strategies with highly evolved

editing tools in a concerted fashion with few to one rounds of edits, assess and optimize recoding efficiencies, followed by massive off-target and bystander mutation cleanup using highly engineered base-editing enzymes.

Please find below a point-by-point response to your comments:

Reviewer #1 (Remarks to the Author):

The authors have responded to most points adequately, and the manuscript has improved significantly on clarity and data presentations. But I think there are still several improvements could be made and a few extra concerns that need to be addressed prior to publication.

1. The figures are much better after the revision. Yet, some are still blurry or misaligned.

Specifically:

- Lots of the graphs, for example figure 1b, figure 2b/c/e/f, figure 3a and figure 4d, are still blurry (the plot, data labels, axis labels, legends) but their axis titles are clear. I assume the authors generated the plots in one software but add the axis titles in another. This problem can be easily solved by exporting the images as vector graphics or just with higher PPI. If it is possible, it would be better to generate all the chart elements in one software so that the font styles and sizes are consistent and coordinated. (Fixed)
- The boxes in figure 2a/d are not aligned and vary in sizes. Also, some annotations, e.g., PSMD13 and NOP2-1, go outside of their boxes. I would suggest adjusting the font sizes to improve the aesthetic. (Fixed)

Thank so much for your suggestion and comments. We have fixed the figures in the revised manuscript.

2. Line 88-106: I agree with reviewer #2 that GRIT is one of the highlights of this manuscript and should be described in more detail. However, I think the readability of this part in the main text would benefit from having more explanations on the design procedure and rules (like those in the supplementary method) instead of getting too bogged down in the details of each python file. Also, the Github link doesn't seem to be working when I try to visit.

Thank you for your comments. We have rewritten the description of GRIT in the main text of the revised manuscript. The software is available now via Github (<https://github.com/thestephencasper/GRIT>). The reason it didn't work before was that it was set to private.

“GRIT offers a toolkit for informatics with an emphasis on recoding. It was created with three key design principles: (1) Portability: all data can be downloaded, and GRIT can be run from a desktop computer in minutes. (2) Adaptability: the full source for the project is in two python files, and a diversity of general and recoding-specific informatics data are readily available including full gene and chromosome sequences. (3) Ease of use: GRIT comes with prewritten methods for replicating results and analyzing chromosome data, gene data, TAG site data, and guides for editing. For recoding in particular, GRIT can be used to index all TAG sites in the genome, search for ones that can be directly edited with a C base editor or edited with a “daisy” chain of A and C editors, and to design the corresponding guides.

GRIT works by creating chromosome and gene objects which each store bioinformatic data. For chromosomes, GRIT gathers data including chromosome name, wildtype sequence, recoded sequence, indices of sites to recode, base editor sites, gene objects, and edit sites that are part of different genes or different codons read in different frames and which two genes they are part of. For genes, GRIT stores the gene name, chromosome, strand, wildtype sequence, recoded sequence, active isoform, introns, isoform information, gene essentiality data, and recoding sites. By making this data readily available, GRIT can be easily adapted for purposes beyond recoding.”

3. Line 147-155: Related to my previous comment #2. Thank you for clarifying the gRNA composition in the gblock pools and the 43-all-in-one construct. To me, this part is very interesting because the authors here demonstrated multiplexed base editing in human cells at a number of different sites that exceeds any previous studies to my knowledge.^{1,2} As the authors mention in the rebuttal, by taking advantages of a stable and inducible CBE cell line, this system overcomes the Cas9 “bottlenecking” problem. I agree with the authors that, within this setting, it is valid to compare gRNA delivery

strategies with increasing number of guides, but I don't think the editing efficiencies of a small collection of 22 common sites, while ignoring 28, 78, and 128 sites respectively, can reflect the true performance of each strategy. With the WES data, the authors have the capability to analyze all the sites and use statistics such as mean, median or total editing efficiencies to represent the performance of each strategy and conduct statistical tests for the comparisons among them. I am curious whether the 20, 30 gBlock pools could have better editing in those non-overlapping sites or could achieve higher total C-to-T conversions. In any case, I presume this analysis would be very informative on determining the theoretical gRNA "ceiling" in HEK cells, and very valuable to the field of base editing.

Thanks so much for your suggestion and comments. We had re-analyzed the WES data and compared the editing efficiency of C-to-T on all mapping regions, the data still show us that 10 gBlocks is more efficient than 20-,30-gBlocks (Supplementary Figure4b, 4c and Table 2).

4. Line 157-161: Related to my comment #3 in the first revision. I appreciate the schematics; it is much clearer now. I would agree with the authors that under such a setup, method_2 is the best. Meanwhile, I notice that the authors put much emphasis on the 43-all-in-one construct as a highlight of this paper. But just like I mentioned previously, it is unfortunate that there is no head-to-head comparison between the 10 gBlock pool strategy and the 43-all-in-one construct, because the method_2 and method_3 use different reporter systems. Given that an activity reporter usually outperforms a transfection reporter, it is still unclear whether or not the 10 gBlock pool strategy is actually better than the 43-all-in-one construct.

Thank you for your comments. We appreciate your agreement with our conclusion that method_2 for implementing the TAG-to-TAA conversion is the best of the three methods in the manuscript. We assess the data based on loci editing efficiency and believe the reporter doesn't affect those conclusions. Besides, the 43-all-in-one construct is the gRNA array that contains the most gRNA reported so far, which was highlighted by us. We agree with you that it is an interesting which method is better

when delivering multiple gRNAs with 10 gBlocks pool and the 43-all-in-one construct, but more experiments are needed to figure out this question in the future, and that is another study.

5. Line 196-197: Related to off-target analysis. I am not sure if clone 1 is the right control for the off-target analysis because it didn't go through the same clonal expansion process as those highly modified clones, especially 19-1, -16 and -21 which were expanded twice from clone 1. It is well documented that clonal expansion leads to genetic diversification.^{3,4} Zuo et al. found 94% of the CBE off-targets are C-to-T mutations, whereas, from a quick calculation, only ~60–80% from this analysis.⁵ Is the high portion of C-to-non-T off targets be a concern of this method? If a better control is not available, I would recommend addressing this issue in the main text.

Thank you for your comments and recommendation. We agree with you that clonal expansion leads to spontaneous somatic mutation and clones with the same clonal expansion process would be better than clone 1 as controls. We considered that clone 1, as the mother cell, also could serve as a control, and we have made annotations in the main text of revised manuscript. Zuo et al. observed off-target SNVs in mouse embryos (diploid cell) upon treatment with BE3 and single sgRNA per sample, but evoBE4max and multiple sgRNAs were used on HEK293T (hypotriploid cell) in our study, which may be difficult to compare directly because of different base editors and cell types. Thus, it's hard to define the high portion of C-to-non-T off targets in our method. In addition, ~25-95% C-to-T of the CBE off-targets were reported in another reference (PMID:31767844). The percentage of C-to-T of the CBE off target maybe different in different experimental conditions.

6. Line 260-267: Related to comment #7 in my previous review. I still think this calculation is not rigorous and too speculative. It seems to oversimplify the task without properly addressing the daunting challenges of recoding the whole genome. In both the rebuttal and the discussion, the authors stated that the off-target problem can be mitigated by using CBE with reduced off targets. However, those CBEs come with

reduced on-target activity as a trade-off -- they won't be as efficient as evo-BE4max used in this paper and it means more rounds of transfection may be needed. Given only 9 homozygous mutations are obtained by using the most efficient CBE available, just recoding 1937 essential genes entails hundreds of rounds of transfection, certainly not "several" rounds as the authors state. In addition, on-target C-to-non-T editing (mostly C-to-G) is never mentioned in this manuscript, which would change a TAG to a TAC encoding tyrosine and result in continued and inappropriate translation of the mRNA into the 3'UTR region. Would the chance of having non-stop mutations be a concern of this method? (BE-HIVE <https://www.crisprbehave.design/> can potentially predict C-to-G editing)

Thank you for your comments. We understand that human recoding is a systematic and complex genome project. Our manuscript firstly describes a workflow for converting TAG to TAA, which will be optimized by recent new editing tool such as DdCBE (PMIDs:32641830,35379961), prime editor (PMIDs:31634902,34653350), Cas12a base editors (PMID:32492431), and so on. The calculation is a preliminary calculation in an ideal case based on multiple sgRNA transfections in parallel and performed separately, and then we picked the better recoded clone for next rounds. This can reduce the number of transfection rounds and experimental time. Recently, some CBE variants carrying less off-target events and high on-target activity have been reported (PMIDs:32345976,32424272). We agree with you that the C-to-non-T editing (mostly C-to-G) is possible which would change a TAG to a TAC encoding tyrosine, and therefore result in continued and inappropriate translation of the mRNA into the 3'UTR region. However, the C-to-non-T editing can induce cytotoxicity, which can lead to cell death. Furthermore, even if there are C-to-non-T editing at some loci, we can also fix them with prime editor in the next round.

7. All in all, I think this paper has already showed impressive data on multiplexed base editing and demonstrated its potential to be further expanded to a larger scale. Thus, in the discussion session, I would recommend spending more efforts to address the roadblocks and challenges so that people in the field can improve upon them.

Thank you for your comments and suggestion. We agree that current approach needs to be further optimized for a larger scale. We have described proposed potential optimization methods (Supplementary Fig.15) during scale-up in the discussion section of the revised manuscript.

“Human recoding is a systematic and complex genome project. Our current study is the initial step of the human genome recoding, which opens the door for subsequent efforts like the E. coli recoding and Sc2.0 synthetic genome. Although we can achieve up to 33 genes in single clones via one transfection so far, we can optimize this framework to scale-up to all essential genes or all genes ending with TAG that can be converted to TAA. Here, we propose several potential strategies: 1) Further developing new base editors with low off-target, high editing efficient and PAM-less/free based on CBE variants^{44, 47, 48}, prime editor^{28, 35} and DdCBE^{46, 49}; 2) Improve sgRNA delivery capability with a larger gRNA array on BAC or YAC vectors and sgRNAs pool; 3) Employ new delivery methods through RNP50 and synchronous transfection (Supplementary Fig.15). The next goal in the human recoding roadmap is to utilize all strategies with highly evolved editing tools in a concerted fashion with few to one rounds of edits, assess and optimize recoding efficiencies, followed by massive off-target and bystander mutation cleanup using highly engineered base-editing enzymes.”

References:

1. McCarty, N. S., Graham, A. E., Studená, L. & Ledesma-Amaro, R. Multiplexed CRISPR technologies for gene editing and transcriptional regulation. Nat. Commun. 11, 1281 (2020).
2. Webber, B. R. et al. Highly efficient multiplex human T cell engineering without double-strand breaks using Cas9 base editors. Nat. Commun. 10, 5222 (2019).
3. Greaves, M. & Maley, C. C. Clonal evolution in cancer. Nature 481, 306–313 (2012).
4. Kakiuchi, N. & Ogawa, S. Clonal expansion in non-cancer tissues. Nat. Rev. Cancer 21, 239–256 (2021).
5. Zuo, E. et al. Cytosine base editor generates substantial off-target single-nucleotide variants in mouse embryos. Science 364, 289–292 (2019).

Reviewer #2 (Remarks to the Author):

In many instances, the authors responded to the reviewers' comments and have improved the manuscript. This study does present some interesting new multiplex genome editing technology in human cells for the very early steps toward recoding the human genome. These advances warrant publication. That said, there are several parts of the manuscript that can still be strengthened. For example, the language in certain parts of the text can be clarified and improved for specificity and accuracy. Also, some figures still appear blurred or distorted and need to be revised before any publication. Some additional comments to address:

Thank you for the insightful suggestion. We have made significant improvements to the images and language throughout the manuscript and believe it is clearer and easier to read.

- Introduction, paragraph 1: the newly revised second sentence is a run-on sentence and should be broken down into multiple sentences that first describe prior work on recoding in bacteria (with citations) followed by how this blueprint can be applied to human recoding and eRF1 engineering. You may want to first describe prior recoding work, as done in the subsequent sentences and paragraphs of the intro, and then transition to how this can be applied to recoding human in the last paragraph of the intro

Thank you for your comments and suggestion. We have rewritten paragraph 1 in the introduction of the revised manuscript.

“Recoding was first established in prokaryote through substitution of the TAG stop codon with TAA and deletion of release factor 1 (RF1)^{2, 8}. Recently, Recoding was implemented genome-wide in E. coli by replacing two sense codons with their synonymous codons, and deleting the corresponding transfer RNA (tRNA)³. Then, Recoding has also been subsequently extended to yeast genome⁹, but its application in the human genome has not been reported so far. Here, we propose human genome recoding to generate virus-resistant cell lines by converting stop codon TAG to TAA, and replacing the endogenous eukaryotic release factor 1 (eRF1) with engineered eRF1

variants (Supplementary Fig. 1a).”

- Results section on GRIT software. I appreciate the authors incorporating more information about this software into the results, but I recommend that the text in the results section describe the design, operations and key results of the software, not the detailed code, which is currently present. Such information should be moved to the methods.

Thank you for your comments. We have rewritten this part in main text of the revised manuscript and moved the detail code to the methods of supplementary data.

“GRIT offers a toolkit for informatics with an emphasis on recoding. It was created with three key design principles: (1) Portability: all data can be downloaded, and GRIT can be run from a desktop computer in minutes. (2) Adaptability: the full source for the project is in two python files, and a diversity of general and recoding-specific informatics data are readily available including full gene and chromosome sequences. (3) Ease of use: GRIT comes with prewritten methods for replicating results and analyzing chromosome data, gene data, TAG site data, and guides for editing. For recoding in particular, GRIT can be used to index all TAG sites in the genome, search for ones that can be directly edited with a C base editor or edited with a “daisy” chain of A and C editors, and to design the corresponding guides.

GRIT works by creating chromosome and gene objects which each store bioinformatic data. For chromosomes, GRIT gathers data including chromosome name, wildtype sequence, recoded sequence, indices of sites to recode, base editor sites, gene objects, and edit sites that are part of different genes or different codons read in different frames and which two genes they are part of. For genes, GRIT stores the gene name, chromosome, strand, wildtype sequence, recoded sequence, active isoform, introns, isoform information, gene essentiality data, and recoding sites. By making this data readily available, GRIT can be easily adapted for purposes beyond recoding.”

- How many instances of alternate isoforms of TAG sites were found in the human genome? This would be interesting information to include.

Thank you for your comments. GRIT identifies 6700 total TAG sites (including ones in

alternate isoforms of the same genes). Of the genes that contain TAG sites, 5266 have one isoform, 574 have two isoforms, 80 have three isoforms, 9 have four isoforms, and 2 have five isoforms. We have also added this information into main text of the revised manuscript.

“Using these data, GRIT identifies 6700 total TAG sites (including ones in alternate isoforms of the same genes). Of the genes that contain TAG sites, 5266 have one isoform, 574 have two isoforms, 80 have three isoforms, 9 have four isoforms, and 2 have five isoforms.”

- Of the 6700 TAG sites, why are 52 of those non-editable by base editors? What do the authors propose as the plan to edit those sites?

Thank you for your comments. There are no PAM sequences available for NG base editor near the TAG codons of these 52 loci. We can edit these 52 TAG sites by using Cas12a base editors (PMID:32492431), prime editor (PMIDs:31634902,34653350), and HDR (PMID:28219395) in the future. We have also added this information into the discussion section of the revised manuscript.

“For TAG codons that cannot be edited by NG-CBE, we plan to edit them by using Cas12a-CBE (PMID:32492431), prime editor (PMIDs:31634902,34653350), and HDR (PMID:28219395) in the future.”

- I cannot find a reference to Fig 1a. I also encourage the authors to enhance visualization for the final publication. (Fixed)
- Fig 1: b: This figure is still distorted and blurred and needs to be corrected before publication; c: (Fixed)
- Fig 2: b, e, and f: title contains non-English characters and the x-axis is cut-off; a & d: lots of different colors and hard to read the font describing each genetic component. Please clarify. (Fixed)

Thank you for your comments. We have added references in Figure1a legend, and improved the quality of figures in the revised manuscript.

- The authors response to both Reviewers' questions regarding the stability of their 30 gblocks does not fully address the question and demonstrate stability. The authors should consider performing experiments that directly investigate this question – both for future work in this project and for others in the field who wish to replicate the multiplexing experiment described in this study. I also think their concluding remarks in response to this question read in a speculative manner.

Thank you for your comments. Regarding the stability of Blocks in plasmid amplifications, we conducted dozens of rounds of transformations of 5-sgRNA array plasmids, then we measured the concentration and sequenced them. We did not observe any mutations at insertion sequences. We also put the sequence alignment results in the supplementary data of revised manuscript (supplementary Fig.3a). For the stability of Blocks in mammalian cells, we transfected gBlock-PC, gBlock-YC1 and the backbone plasmid as control with 20ng EGFP plasmid as transfection marker into HEK293T clone 1 separately. We detected transfection efficiency by FACS and the copy number of the three plasmids at 12h, 24h, 48h and 72h after transfection by qPCR. The data show that transfection efficiency among the three plasmids had no significant differences (supplementary Fig.3b). The copy number of the three plasmids was decreased with the increases in transfection time, but no significant differences between the three plasmids' copy number at each time point (supplementary Fig.3c). These data demonstrate gBlocks are stable during plasmid amplification and transfection into mammalian cells.

Reviewers' Comments:

Reviewer #1:

Remarks to the Author:

the authors have done an excellent job revising their manuscript to respond to my comments. I have no additional concerns.